# SPECULATIVE RAG: ENHANCING RETRIEVAL AUGMENTED GENERATION THROUGH DRAFTING

**Zilong Wang**[1]* **Zifeng Wang**[2] **Long T. Le**[2] **Huaixiu Steven Zheng**[3]
**Swaroop Mishra**[3] **Vincent Perot**[3] **Yuwei Zhang**[1] **Anush Mattapalli**[4]
**Ankur Taly**[4] **Jingbo Shang**[1] **Chen-Yu Lee**[2] **Tomas Pfister**[2]
[1]University of California, San Diego  [2]Google Cloud AI Research
[3]Google DeepMind  [4]Google Cloud AI

## ABSTRACT

Retrieval augmented generation (RAG) combines the generative abilities of large language models (LLMs) with external knowledge sources to provide more accurate and up-to-date responses. Recent RAG advancements focus on improving retrieval outcomes through iterative LLM refinement or self-critique capabilities acquired through additional instruction tuning of LLMs. In this work, we introduce SPECULATIVE RAG – a framework that leverages a larger generalist LM to efficiently verify multiple RAG drafts produced in parallel by a smaller, distilled specialist LM. Each draft is generated from a distinct subset of retrieved documents, offering diverse perspectives on the evidence **while reducing input token counts per draft**. This approach enhances comprehension of each subset and mitigates potential **position bias over long context**. Our method accelerates RAG by delegating drafting to the smaller specialist LM, with the larger generalist LM performing a **single** verification pass over the drafts. Extensive experiments demonstrate that SPECULATIVE RAG achieves **state-of-the-art performance with reduced latency** on TriviaQA, MuSiQue, PopQA, PubHealth, and ARC-Challenge benchmarks. It notably enhances accuracy by up to 12.97% while reducing latency by 50.83% compared to conventional RAG systems on PubHealth.

## 1 INTRODUCTION

Large language models (LLMs) have demonstrated remarkable success in question answering tasks (Brown et al., 2020; Achiam et al., 2023; Team et al., 2023). Trained on massive datasets, LLMs leverage their extensive parametric memory to generate seemingly plausible responses to user queries (Kojima et al., 2022; Kamalloo et al., 2023). However, when faced with knowledge-intensive questions demanding up-to-date information or obscure facts (Petroni et al., 2021), LLMs struggle with factual inaccuracies and produce hallucinated contents (Huang et al., 2023).

Retrieval Augmented Generation (RAG) has emerged as a promising solution to mitigate these issues. By incorporating information retrieved from an external database into the context (Gao et al., 2023b), RAG effectively reduces factual errors in knowledge-intensive tasks. This approach not only enables easy and efficient access to vast databases but also facilitates timely and accurate knowledge integration. Due to the inherent limitations in the precision of current dense retrievers and the vastness of knowledge required to answer complex questions (Chen et al., 2022), RAG systems typically retrieve multiple documents to ensure the inclusion of all necessary information in the context (Petroni et al., 2021). This practice inevitably increases the length of the input to the LLMs, presenting significant challenges, particularly since encoding lengthy retrieved documents incurs additional latency and require more complex reasoning. Recent studies have explored ways to extend the context length limit of LLMs (Ding et al., 2023; Reid et al., 2024; Ma et al., 2024), yet achieving well-grounded reasoning over extended contexts remains an open question (Liu et al., 2024; Li et al., 2024). Consequently, striking a balance between efficiency and effectiveness in RAG has become a central research question in the literature. Existing work on RAG systems primarily

---

*Work done while the author was a student researcher at Google Cloud AI Research. Correspondence to: Zilong Wang <zlwang@ucsd.edu>, Chen-Yu Lee <chenyulee@google.com>

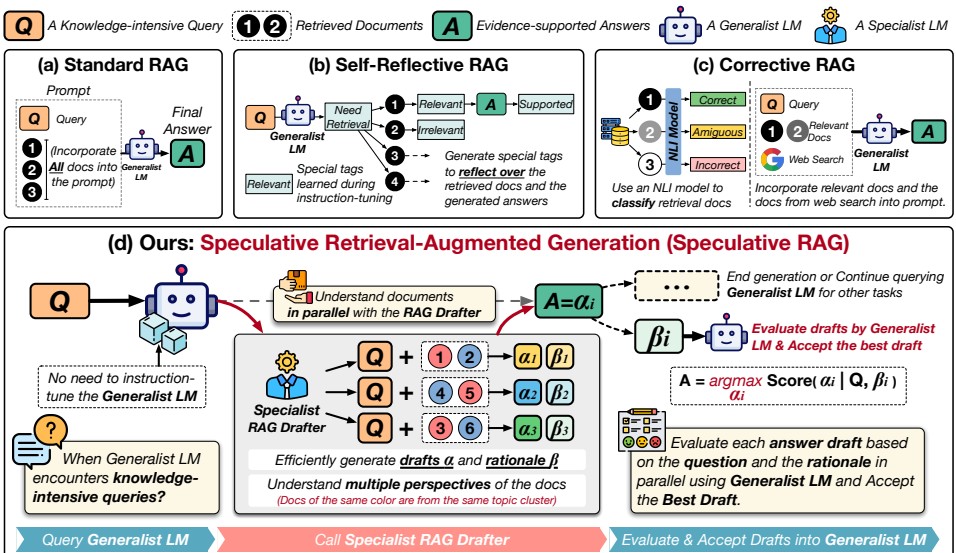

Figure 1: Illustration of different RAG approaches. Given a knowledge-intensive query $Q$ and retrieved documents, (a) Standard RAG incorporates all documents into the prompt, increasing input length and slowing inference; (b) Self-Reflective RAG (Asai et al., 2023) requires specialized instruction-tuning of the general-purpose language model (LM) to generate specific tags for self-reflection; (c) Corrective RAG (Yan et al., 2024) employs an external retrieval evaluator to refine document quality, focusing solely on contextual information without enhancing reasoning capabilities; (d) In contrast, our proposed SPECULATIVE RAG leverages a larger generalist LM to efficiently verify multiple RAG drafts produced in parallel by a smaller, specialized LM. Each draft is generated from a distinct subset of retrieved documents, providing diverse perspectives on the evidence while minimizing the number of input tokens per draft.

concentrates on improving the quality of contextual information in retrieval outcomes, but often neglecting the latency issues associated with these systems (Ma et al., 2023; Baek et al., 2023; Yan et al., 2024; Xie et al., 2023; Asai et al., 2023; Feng et al., 2023). These methods typically rely on multiple refinement iterations or customized instruction-tuning for self-critique abilities. Integrating such enhancements into generic LMs requires additional training or increased latency, posing practical challenges in real-world applications.

To this end, we introduce SPECULATIVE RAG, a RAG framework designed to offload computational burden to a smaller, specialist LM that serves as an efficient and robust RAG module for existing generalist LMs. Inspired by Speculative Decoding (Leviathan et al., 2023; Chen et al., 2023a; Xia et al., 2024a), which accelerates auto-regressive LM inference by concurrently generating multiple draft tokens with a smaller model and verifying them in parallel with the base model, our approach adapts this concept to RAG.

In SPECULATIVE RAG, we partition retrieved documents into subsets for drafting answer candidates. We cluster the retrieved documents by content similarity and sample one document from each cluster to form a subset, minimizing redundancy and maximizing diversity. These document subsets are then fed to multiple instances of the RAG module, which generate draft answers with corresponding rationales in parallel. This smaller, specialized RAG module, excels at reasoning over retrieved documents and can rapidly produce accurate responses. Subsequently, the generalist LM bypasses the detailed review of potentially repetitive documents, focusing instead on validating the drafts against the rationales to determine the most accurate answer. We utilize the strong language modeling capabilities of generalist LMs, calculating the conditional generation probability of the answer drafts and rationales as a confidence score. Our key contributions are:

- We introduce a novel RAG framework that employs a smaller specialist RAG drafter to generate high-quality draft answers. Each draft is derived from a distinct subset of retrieved documents, offering diverse perspectives while reducing input token counts per draft.
- The generalist LM, operating with the RAG drafter, requires no additional tuning. It simply verifies and integrates the most promising draft into the final answer. This approach enhances

comprehension of each subset and mitigates potential lost-in-the-middle (Liu et al., 2024) phenomenon.

- Our method significantly accelerates RAG by delegating drafting to the smaller specialist LM, with the larger generalist LM performing a single, unbiased verification pass over the drafts in parallel. Extensive experiments on 5 free-form question-answering and closed-set generation benchmarks demonstrate the superior effectiveness and efficiency of the method.

## 2 RELATED WORKS

**Retrieval Augmented Generation** Retrieval Augmented Generation (RAG) enhances LLMs by retrieving relevant documents from external databases and incorporating them into the generation process (Gao et al., 2023b; Lewis et al., 2020; Khandelwal et al., 2020; Izacard & Grave, 2021; Luo et al., 2023a; Xia et al., 2024b; Wang et al., 2024). Recent work has primarily focused on enabling LLMs to understand when and what to retrieve (Ma et al., 2023; Chen et al., 2023b; Jiang et al., 2023b; Schick et al., 2024), or designing approaches to better utilize contexts (Yu et al., 2023; Yoran et al., 2023; Wang et al., 2023b; Sarthi et al., 2024; Baek et al., 2023; Xu et al., 2023; Kim et al., 2024). Among them, SAIL (Luo et al., 2023a) fine-tunes a pre-trained LLM on web search data to filter irrelevant contents. Self-Reflective RAG (Asai et al., 2023) introduces reflection tokens to guide retrieval and annotation in instruction-tuning datasets. However, both approaches require additional instruction-tuning of generic LLMs, which is resource-intensive and may lead to forgetting or over-fitting (Luo et al., 2023b). Furthermore, long context with retrieved documents can suffer from computational inefficiency and position bias (Liu et al., 2024). Corrective RAG (Yan et al., 2024) on the other hand proposes a lightweight retrieval evaluator, but it lacks the capability for high-level reasoning. In contrast, our proposed SPECULATIVE RAG addresses these limitations by leveraging a smaller RAG drafter model to efficiently understand diverse perspectives in retrieval results and generate drafts for the generalist LMs to verify and integrate.

**Speculative Decoding** Speculative decoding (Stern et al., 2018; Xia et al., 2023; Chen et al., 2023a; Leviathan et al., 2023; Xia et al., 2024a) aims to reduce auto-regressive decoding latency through a draft-then-verify paradigm. This involves drafting multiple future tokens with a small model and verifying them in parallel with the target model (Xia et al., 2024a). The draft model is typically either an independent model from the same series (Leviathan et al., 2023; Chen et al., 2023a) or the target model itself (Zhang et al., 2023a; Cai et al., 2024). Our approach extends this concept from token-level drafting to answer-level drafting. In contrast to traditional verification criteria (Stern et al., 2018; Xia et al., 2023; Leviathan et al., 2023; Chen et al., 2023a; Miao et al., 2024), which accept or reject tokens based on their generation probabilities, we leverage language modeling objectives to directly assess the confidence of entire answer drafts.

## 3 SPECULATIVE RETRIEVAL AUGMENTED GENERATION THROUGH DRAFTING

**Problem Formulation** In knowledge intensive tasks, each entry can be represented as $(Q, D, A)$, where $Q$ is a question or statement that requires additional knowledge; $D = \{d_1, ..., d_n\}$ is a set of $n$ documents retrieved from the database; $A$ is the expected answer. Particularly, in question answering tasks, $Q$ and $A$ are the question and the expected answer in natural language form; in the statement verification tasks, $Q$ is a statement and $A \in \{\text{True}, \text{False}\}$ is a Boolean value indicating the statement's correctness; in the multiple choice tasks, $Q$ is a question with a few options and $A \in \{\text{A}, \text{B}, \text{C}, ...\}$ is the index of the correct answer. The objective of a RAG system is to generate a fluent response containing the expected answer or select the expected answer from the provided options based on the context provided by the retrieved supporting documents.

### 3.1 OVERVIEW

We introduce Speculative Retrieval Augmented Generation (SPECULATIVE RAG), as illustrated in Figure 1. We aim at enhancing the reasoning ability of LLMs over retrieved documents without compromising processing speed. Instead of relying on brute-force parameter scaling or instruction-tuning an entire LM to handle knowledge-intensive tasks, we propose a divide-and-conquer approach. We utilize **a smaller specialist LM**, **the RAG drafter**, to rapidly generate multiple answer

drafts based on retrieved results. Then, **a larger generalist LM**, **the RAG verifier**, assesses these drafts, selects the best one based on its rationale, and integrates it into the generation results.

---

**Algorithm 1:** SPECULATIVE RAG

    **Data:** $(Q, D = \{d_i\}_i^n)$ is the question and $n$ retrieved documents; $m$ subsets, each containing $k$ documents, are sampled from $D$; $k$ also corresponds to the number of clusters during clustering.

    **Result:** $\hat{A}$ is the predicted answer to the question.

**1**  **Function** Speculative RAG $(Q, D, m, k)$:

**2**     $\{c_1, c_2, ..., c_k\} \xleftarrow{\text{K-Means}} \mathcal{C}(d_1, ..., d_n | Q)$     ▷ Cluster the documents into $k$ groups using an embedding model $\mathcal{C}$.

**3**     $\Delta \leftarrow \{\}$

**4**     **repeat**

**5**         $\boldsymbol{\delta}_j \leftarrow \{\}$     ▷ Construct a subset of the retrieved documents $\boldsymbol{\delta}_j$

**6**         **for** $c_i \in \{c_1, ..., c_k\}$ **do**

**7**             $\boldsymbol{\delta}_j = \boldsymbol{\delta}_j \cup \{\text{random.sample}(c_i)\}$     ▷ Sample one document from each cluster $c_i$ into subset $\boldsymbol{\delta}_j$.

**8**         **end**

**9**         $\Delta = \Delta \cup \{\boldsymbol{\delta}_j\}$

**10**     **until** $|\Delta| = m$     ▷ Repeat the sampling until there are $m$ unique subsets in total.

**11**     **for** $\boldsymbol{\delta}_j \in \Delta$ **do** *in parallel*     ▷ Process $m$ subsets in parallel.

**12**         $\alpha_j, \beta_j \leftarrow \mathcal{M}_{\text{Drafter}}.\text{generate}(Q, \boldsymbol{\delta}_j)$     ▷ Generate the **draft** $\alpha$ and **rationale** $\beta$ with $\mathcal{M}_{\text{Drafter}}$.

**13**         $\rho_j \leftarrow \mathcal{M}_{\text{Verifier}}.\text{score}(\alpha_j | Q, \beta_j)$     ▷ Compute the confidence score $\rho$ with $\mathcal{M}_{\text{Verifier}}$.

**14**     **end**

**15**     $\hat{A} \leftarrow \arg\max_{\alpha_j} \rho_j$     ▷ Select the one with the highest score as the final answer.

**16**  **return** $\hat{A}$

---

Specifically, as shown in Algorithm 1, we first cluster the retrieved documents with regard to their relation to the posed question, where each cluster represents one perspective in the retrieval results (Line 2). Then we sample one document from each cluster into a subset so the documents in this subset covers the multiple perspectives in the retrieval results. We aim at minimizing redundancy and increase the diversity of the documents (Line 5 to 8). We denote one subset as $\boldsymbol{\delta} \subset D$ that contains retrieved documents with diverse contents and multiple perspectives in the retrieval results. Then, we distribute each subset $\boldsymbol{\delta}$ to a RAG drafter endpoint $\mathcal{M}_{\text{Drafter}}$ with the posed question $Q$ to generate the answer draft $\alpha$ and the rationale $\beta$ in parallel (Line 12). The RAG drafter is instruction-tuned to be a specialist in understanding the retrieved documents and produce rationales that are faithful to the input documents. It is smaller than generalist LMs, and its parallel processing further ensures high efficiency. For each draft-rationale pair $(\alpha, \beta)$ from $\mathcal{M}_{\text{Drafter}}$, we compute a confidence score with the generalist LM $\mathcal{M}_{\text{Verifier}}$ based on the question $Q$ and corresponding rationale $\beta$ (Line 13). It is worth mentioning that $\mathcal{M}_{\text{Verifier}}$ does not need to be instruction-tuned since we leverage its language modeling ability already learned during pre-training. Meanwhile, $\mathcal{M}_{\text{Verifier}}$ can verify the drafts based on the informative rationale provided by $\mathcal{M}_{\text{Drafter}}$ instead of processing tedious or possibly redundant retrieved documents. Finally, we select the answer draft with the highest confidence score as the final answer and integrate it into the generation results of the generalist LM (Line 15).

## 3.2 SPECIALIST RAG DRAFTER

Instead of tuning a large generalist LM for the RAG scenario, we leverage a smaller specialist LM, $\mathcal{M}_{\text{Drafter}}$, to understand retrieved documents. $\mathcal{M}_{\text{Drafter}}$ is specialized in answering the given question based on the supporting documents and not expected to cope with general problems. It serves as a RAG module for the generalist LMs when solving knowledge-intensive tasks. We train $\mathcal{M}_{\text{Drafter}}$ to generate both the answer draft and the rationale to better understand the contextual documents.

**Instruction Tuning** Given a triplet $(Q, A, D)$, where $Q$ is a general query, $A$ is the response, and $D$ is a retrieved supporting document, we augment it with the rationale of the response $A$ based on the document $D$. We denote the rationale as $E$ which extracts essential information from the document and explains why the response is reasonable to the query concisely (Hsieh et al., 2023) so it is of shorter length and delivers information coherent with the original document. We leverage relatively strong LMs to automatically synthesize the rationale $E$ for each triplet. Specifically, we directly query the strong LM to understand the knowledge from the document and provide the intermediate rationale between the instruction and response. Refer to Appendix G for detailed prompts. After generating the rationale, we finetune a pre-trained LM using the standard language modeling objective, maximizing the likelihood: $\mathbb{E}_{(Q,A,D,E)} \log P_{\mathcal{M}_{\text{Drafter}}}(A, E \mid Q, D)$, where $(Q, A, D, E)$ is an

augmented entry in the dataset; $P_{\mathcal{M}_{\text{Drafter}}}(A, E \mid Q, D)$ is the probability of generating the response and rationale based on the query and document. We use this instruction-tuned model as the specialist RAG drafter which learns to generate a well-grounded response and rationale given the query and relevant documents.

**Multi-Perspective Sampling** For each knowledge-intensive question, we retrieve a set of documents from the database using the posed question as the retrieval query. These documents may contain diverse content due to the ambiguity inherent in the query. To minimize redundancy and enhance diversity of the document subsets used for generating answer drafts, we employ a multi-perspective sampling strategy. We first cluster the documents into a few topics using an instruction-aware embedding model (Peng et al., 2024) and the K-Means clustering (Jin & Han, 2011).

$$\text{emb}(d_1), ..., \text{emb}(d_n) = \mathcal{E}(d_1, ..., d_n | Q)$$
$$\{c_1, ..., c_k\} = \text{K-Means}(\text{emb}(d_1), ..., \text{emb}(d_n))$$
$$\delta = \{\text{random.sample}(c) \text{ for } c \in \{c_i\}_1^k\}$$

where $\mathcal{E}$ is an instruction-aware embedding model which embeds a string with regard to a provided instruction (the posed question $Q$); $\text{emb}(d_i)$ is the embedding for the retrieved document $d_i$; $c_j$ is a cluster of retrieved documents with similar topics and contents; $k$ is a hyper-parameter that controls the number of clusters. We sample one document from each cluster into a document subset $\delta$ so each subset contains $k$ documents of diverse contents. In total, we construct $m$ subsets for parallel inference with the RAG drafter.

**RAG Drafting** We run $\mathcal{M}_{\text{Drafter}}$ over the $m$ document subsets and produce corresponding answer drafts. Refer to Appendix H for detailed prompt. We incorporate each document subset into the prompt and query $\mathcal{M}_{\text{Drafter}}$ for responses. We obtain $m$ drafts as the answer candidates and each draft is grounded based on the multiple perspectives in the retrieval results. Specifically, given a document subset $\delta_j = \{d_{j_1}, .., d_{j_k}\}$, we query $\mathcal{M}_{\text{Drafter}}$ in parallel with the following prompt for the answer draft and rationale: $Q, d_{j_1}, ..., d_{j_k} \rightarrow \alpha_j, \beta_j$, where the prompt contains the posed question $Q$ along with the document subset; the generation result contains the answer draft $\alpha$ and the rationale $\beta$. We denote the conditional generation probability as $\rho_{\text{Draft},j} = P(\beta_j | Q, d_{j_1}, ..., d_{j_k}) + P(\alpha_j | Q, d_{j_1}, ..., d_{j_k}, \beta_j)$, which measures the reliability of generating rationales and the confidence in producing answer drafts.

## 3.3 GENERALIST RAG VERIFIER

After generating drafts and the rationale from the RAG drafter $\mathcal{M}_{\text{Drafter}}$, we evaluate them by a generalist LM $\mathcal{M}_{\text{Verifier}}$ to filter out the less reliable drafts and select the best answer. The generalist LM can be any off-the-shelf pre-trained LM. We only consider the draft-rationale pair $(\alpha, \beta)$ and skip the tedious and redundant retrieval results. We resort to the language modeling ability of the generalist LM to rank and select the draft-rationale pairs.

**Evaluation Scores** First, we calculate the **self-consistency score** by determining the conditional probability of generating a draft-rationale pair given the question, $\rho_{\text{Self-contain}} = P(\alpha, \beta | Q)$. This score helps assess whether the draft and rationale are self-consistent in the context of the question. Given the characteristics of language modeling, a self-consistent draft-rationale pair is expected to yield a higher probability. Furthermore, we incorporate a self-reflection statement $R$ that prompts $\mathcal{M}_{\text{Verifier}}$ to assess the reliability of an answer draft (e.g. "Do you think the rationale supports the answer, yes or no?"). We define the **self-reflection score** as $\rho_{\text{Self-reflect}} = P(\text{"Yes"} | Q, \alpha, \beta, R)$ where we compute the conditional probability of the positive answer ("Yes") to the self-reflection statement.

**Computation Method** We can efficiently compute the self-consistency and self-reflection scores within one forward pass of $\mathcal{M}_{\text{Verifier}}$. Given a question $Q$ and a draft-rationale pair $(\alpha, \beta)$, we construct a prompt $[Q, \alpha, \beta, R, \text{"Yes"}]$, where $R$ is the self-reflection statement. We encode the prompt with $\mathcal{M}_{\text{Verifier}}$, and acquire the probability of each token conditioned on the previous tokens $P(t_i | t_{<i})$. We leverage this auto-regressive feature and aggregate the probability of the relevant tokens to compute the self-consistent score $\rho_{\text{Self-contain}}$ and self-reflection score $\rho_{\text{Self-reflect}}$.

$$Q, \overbrace{\alpha, \beta}^{\rho_{\text{SC}}}, R, \overbrace{\text{"Yes"}}^{\rho_{\text{SR}}} \Rightarrow \begin{cases} \rho_{\text{SC}} = \prod_{t_i \in \alpha} P(t_i | t_{<i}) \cdot \prod_{t_i \in \beta} P(t_i | t_{<i}) \\ \rho_{\text{SR}} = \prod_{t_i \in \text{"Yes"}} P(t_i | t_{<i}) \end{cases}$$

Finally, we produce the final score, $\rho_j = \rho_{\text{Draft},j} \cdot \rho_{\text{SC},j} \cdot \rho_{\text{SR},j}$, and then select the most reliable answer as the final answer to the question $\hat{A} = \arg\max_{\alpha_j} \rho_j$.

# 4 EXPERIMENTS

We evaluate our proposed SPECULATIVE RAG on five public retrieval augmented generation benchmarks: TriviaQA (unfiltered) (Joshi et al., 2017), MuSiQue (Trivedi et al., 2022), PopQA (Mallen et al., 2023), PubHealth (Zhang et al., 2023b), and ARC-Challenge (Clark et al., 2018). We provide representative examples for case study in Appendix J. TriviaQA, MuSiQue, PopQA are challenging open-domain question answering datasets where RAG systems are required to answer questions on factual knowledge. TriviaQA and PopQA typically require one accurate piece of evidence from the documents, whereas MuSiQue demands multiple documents to construct a multi-hop reasoning chain. More detailed experiments on multi-hop reasoning can be found in Appendix F. Following previous works (Guu et al., 2020; Asai et al., 2023; Yan et al., 2024), we evaluate performance of the free-form generation based on whether gold answers are contained within the generated response or not. PubHealth and ARC-Challenge are closed-set generation datasets. PubHealth is a dataset of medical claims spanning a variety of biomedical subjects and it requires the RAG system to verify a given claim based on the retrieved documents. ARC-Challenge introduces a multi-choice question answering dataset, composed of science exam questions from grade 3 to grade 9. For closed-set generation tasks, we use accuracy to evaluate whether the generated answers match the ground truth.

## 4.1 BASELINES

**Standard RAG**   For standard RAG, we incorporate all the retrieved documents into the prompt as contextual information. Refer to Appendix I for detailed prompts. We run standard RAG experiments on off-the-shelf LLMs including Mistral$_{7B}$, Mistral-Instruct$_{7B}$ (Jiang et al., 2023a), Mixtral$_{8x7B}$, Mixtral-Instruct$_{8x7B}$ (Jiang et al., 2024), and Alpaca$_{7B}$ (Dubois et al., 2024). We also include the performance of Toolformer (Schick et al., 2024) and SAIL (Luo et al., 2023a) which are originally reported from Asai et al. (2023). Toolformer$_{7B}$ is an LM instruction-tuned to use tools including a search engine, and SAIL$_{7B}$ is an LM instruction-tuned on the Alpaca instruction tuning set augmented with search results from different sources such as DuckDuckGo and Wikipedia.

**Self-Reflective RAG and Corrective RAG**   Self-Reflective RAG (Self-RAG)  (Asai et al., 2023) and Corrective RAG (CRAG)  (Yan et al., 2024) are more advanced RAG systems that enhances the quality of contextual information in the retrieval results. CRAG introduces an external evaluator to assess the quality of retrieved documents, and to refine them before the response generation. Self-RAG instruction-tunes an LM to generate special self-refection tags. These tags guides the LM to dynamically retrieve documents when necessary, critique the retrieved documents relevance before generating responses. Self-CRAG is to apply the Self-RAG approach on the refined documents of CRAG. We adopt the same backbone LLMs across all methods as our proposed SPECULATIVE RAG for fair comparisons.

## 4.2 EXPERIMENT SETTINGS

In our experiments, we utilize Mistral$_{7B}$ (v0.1) as our base LM for the RAG drafter. For RAG verifier, we employ either Mistral$_{7B}$ (v0.1) or Mixtral$_{8x7B}$ (v0.1) without any fine-tuning, denoted as $\mathcal{M}_{\text{Verifier-7B}}$ or $\mathcal{M}_{\text{Verifier-8x7B}}$. We pre-compute embeddings of retrieved documents using a lightweight instruction-aware embedding model InBedder$_{\text{Roberta}}$ (Peng et al., 2024) as part of the retrieval process. Inference is conducted using the vLLM framework (Kwon et al., 2023) with greedy decoding (temperature = 0). We adopt the same experiment settings from Asai et al. (2023) and include a more challenging benchmark, MuSiQue (Trivedi et al., 2022). Our focus is on RAG reasoning rather than evidence citation, so we omit the other two long-form generation benchmarks, Biography (Min et al., 2023) and ALCE-ASQA (Gao et al., 2023a). On TriviaQA, PopQA, PubHealth, and ARC-Challenge, we retrieve top 10 documents and generate 5 drafts per query ($m = 5$), with each draft based on a subset of 2 documents ($k = 2$). For MuSiQue, we retrieve top 15 documents and generate 10 drafts for each query ($m = 10$), each using a subset of 6 documents due to more complex reasoning. Further details regarding instruction-tuning can be found in Appendix A.

Table 1: Retrieval augmentation generation results on TriviaQA, MuSiQue, PopQA, PubHealth, and ARC-Challenge (ARC-C). (*We use the RAG drafter's generation probability $\rho_{Draft}$ as the confidence score for selecting drafts when we use it alone; [†] indicates numbers reported in Asai et al. (2023); − denotes numbers that are not reported by the original papers or are not applicable; [‡]we use Mistral$_{7B}$ or Mixtral$_{8x7B}$ as the RAG verifier, and denote them as $\mathcal{M}_{\text{Verifier-7B}}$ or $\mathcal{M}_{\text{Verifier-8x7B}}$.)

| RAG Method | Free-form | | | Closed-set | |
| --- | --- | --- | --- | --- | --- |
| | TriviaQA | MuSiQue | PopQA | PubHealth | ARC-C |
| *Standard RAG* | | | | | |
| Mistral$_{7B}$ (Jiang et al., 2023a) | 54.15 | 16.71 | 31.38 | 34.85 | 42.75 |
| Mixtral$_{8x7B}$ (Jiang et al., 2024) | 59.85 | 19.16 | 34.02 | 37.08 | 48.72 |
| Mistral-Instruct$_{7B}$ (Jiang et al., 2023a) | 67.11 | 17.99 | 42.17 | 42.15 | 47.70 |
| Mixtral-Instruct$_{8x7B}$ (Jiang et al., 2024) | 73.91 | 29.42 | 53.68 | 63.63 | 78.41 |
| Alpaca$_{7B}$ (Dubois et al., 2024)[†] | 64.1 | - | 46.7 | 40.2 | 48.1 |
| Toolformer$_{6B}$ (Schick et al., 2024)[†] | 48.8 | - | - | - | - |
| SAIL$_{7B}$ (Luo et al., 2023a)[†] | - | - | - | 69.2 | 48.4 |
| *Self-Reflective RAG & Corrective RAG* | | | | | |
| CRAG$_{\text{Mistral-7B}}$ (Yan et al., 2024) | 59.03 | - | 49.46 | 59.04 | 74.87 |
| Self-RAG$_{\text{Mistral-7B}}$ (Asai et al., 2023) | 64.84 | 21.72 | 52.68 | 72.44 | 74.91 |
| Self-CRAG$_{\text{Mistral-7B}}$ (Yan et al., 2024) | 65.43 | - | 56.11 | 72.85 | 75.26 |
| ***Our Speculative RAG*** | | | | | |
| $\mathcal{M}_{\text{Drafter-7B}}$* | 71.11 | 27.89 | 56.40 | 75.58 | 74.49 |
| $\mathcal{M}_{\text{Verifier-7B}}$[‡] + $\mathcal{M}_{\text{Drafter-7B}}$ | 73.91 | 31.03 | 56.75 | 75.79 | 76.19 |
| $\mathcal{M}_{\text{Verifier-8x7B}}$[‡] + $\mathcal{M}_{\text{Drafter-7B}}$ | **74.24** | **31.57** | **57.54** | **76.60** | **80.55** |

## 4.3 MAIN RESULTS

We compare SPECULATIVE RAG with standard RAG approaches, as well as the more advanced Self-Reflective RAG and Corrective RAG on five datasets: TriviaQA, MuSiQue, PopQA, PubHealth, and ARC-Challenge. We report the performance of $\mathcal{M}_{\text{Drafter-7B}}$ when used alone or paired with the RAG verifier (e.g. $\mathcal{M}_{\text{Verifier-7B}}$, $\mathcal{M}_{\text{Verifier-8x7B}}$). Following prior work (Asai et al., 2023; Yan et al., 2024), we report accuracy as the performance metric.

**Superior Performance over Baselines** Table 1 demonstrates that our method consistently outperforms all baselines across all five benchmarks. Particularly, $\mathcal{M}_{\text{Verifier-8x7B}}$ + $\mathcal{M}_{\text{Drafter-7B}}$ surpasses the most competitive standard RAG model, Mixtral-Instruct$_{8x7B}$, by 0.33% on TriviaQA, 2.15% on MuSiQue, 3.86% on PopQA, 12.97% on PubHealth, and 2.14% on ARC-Challenge. With a comparable number of instruction-tuned parameters, $\mathcal{M}_{\text{Verifier-7B}}$ + $\mathcal{M}_{\text{Drafter-7B}}$ outperforms all Self-Reflective and Corrective RAG methods, and $\mathcal{M}_{\text{Drafter}}$ alone surpasses the baselines in most settings.

**Effective Instruction Tuning for RAG Drafter** Our instruction tuning is effective in enhancing the reasoning ability of the drafter model (Hsieh et al., 2023), as we observe a remarkable performance improvement comparing Mistral$_{7B}$ and $\mathcal{M}_{\text{Drafter-7B}}$. Additionally, we further investigate the performance of $\mathcal{M}_{\text{Drafter-7B}}$ when we directly feed all documents to the RAG drafter and generate one draft, with detailed results provided in Appendix B. Moreover, the performance of Mixtral$_{8x7B}$ significantly improves when paired with the instruction-tuned RAG drafter $\mathcal{M}_{\text{Drafter-7B}}$, showing gains of 14.39% on TriviaQA, 12.41% on MuSiQue, 23.52% on PopQA, 39.52% on PubHealth, and 31.83% on ARC-Challenge. Similar improvements are observed with Mistral$_{7B}$ as well. For Mistral$_{7B}$, we observed improvements of 19.76% on TriviaQA, 14.32% on MuSiQue, 25.37% on PopQA, 40.94% on PubHealth, and 33.44% on ARC-Challenge. We attribute these improvements to the superior reasoning capabilities of the RAG drafter over the retrieved documents in SPECULATIVE RAG. By minimizing the redundancy in the sampled documents, the RAG drafter generates higher quality answer drafts based on diverse perspectives from the retrieval results.

**Reliable Scoring by RAG Verifier** The reliable draft verification by the generalist LM also contributes to the enhanced performance. The performance improves remarkably comparing $\mathcal{M}_{\text{Drafter-7B}}$ and $\mathcal{M}_{\text{Verifier-7B}}$ + $\mathcal{M}_{\text{Drafter-7B}}$. The instruction-tuned RAG drafter is specialized in generating answer drafts based on the retrieved documents while the language modeling capabilities of generic LMs are leveraged to validate each draft in light of its rationale. This method is both effective and easy to implement, showcasing the effectiveness of this verification approach.

## 4.4 Effects of Generated Rationale for Verification

In Speculative RAG, we utilize the generated rationale $\beta$ from the RAG drafter as an indicator of the trustworthiness of answer drafts $\alpha$.

**Shortened length compared to the retrieved documents.** The rationales highlight relevant points, omit redundant information, and bridge logical gaps between drafts and their supporting documents. We compare the number of tokens in the generated rationale and the retrieved documents, and plot them in Figure 2. We find that the generated rationale is significantly shorter than the retrieved documents.

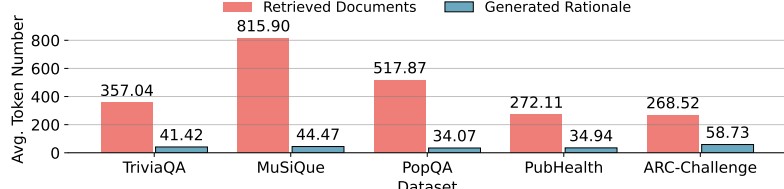

Figure 2: Average number of tokens in the generated rationale and the retrieved documents in TriviaQA, MuSiQue, PopQA, PubHealth, and ARC-Challenge. The generated rationale is of much shorter length than the original retrieved documents.

Table 2: Performance and latency analysis of Speculative RAG on TriviaQA and PubHealth using $\mathcal{M}_{\text{Verifier-8x7B}} + \mathcal{M}_{\text{Drafter-7B}}$. We add the original document subset $\boldsymbol{\delta}$ to the context or replace the generated rationale $\beta$ with the original retrieved document subset $\boldsymbol{\delta}$ during verification, i.e. we compute the self-containment score as $\rho_{\text{Self-contain}} = P(\alpha, \boldsymbol{\delta}|Q)$ or $\rho_{\text{Self-contain}} = P(\alpha, \boldsymbol{\delta}, \beta|Q)$, and compute the self-reflection score as $\rho_{\text{Self-reflect}} = P(\texttt{"Yes"}|Q, \alpha, \boldsymbol{\delta}, R)$ or $\rho_{\text{Self-reflect}} = P(\texttt{"Yes"}|Q, \alpha, \boldsymbol{\delta}, \beta, R)$, where $Q$ is the query; $\alpha$ is the answer draft; $R$ is the self-reflection statement.

| | TriviaQA | | PubHealth | |
|---|---|---|---|---|
| | **Accuracy (%)** | **Latency (s)** | **Accuracy (%)** | **Latency (s)** |
| $\mathcal{M}_{\text{Verifier-8x7B}} + \mathcal{M}_{\text{Drafter-7B}}$ | | | | |
| $\rho = \texttt{Score}(\alpha\|Q, \beta)$ | 74.24 | **1.93** | **76.60** | **1.17** |
| $\rho = \texttt{Score}(\alpha\|Q, \boldsymbol{\delta})$ | 74.08 (-0.16) | 2.13 (+10.36%) | 76.09 (-0.51) | 1.31 (+11.97%) |
| $\rho = \texttt{Score}(\alpha\|Q, \beta, \boldsymbol{\delta})$ | **74.32** (+0.08) | 2.17 (+12.44%) | 76.29 (-0.31) | 1.33 (+13.68%) |

**Comparable performance with retrieved documents and lower latency.** To evaluate the effectiveness of the rationales, we create two alternative scoring methods: (a) replacing rationale with retrieved documents ($\rho = \texttt{Score}(\alpha|Q, \boldsymbol{\delta})$), or (b) adding retrieved documents to rationale ($\rho = \texttt{Score}(\alpha|Q, \beta, \boldsymbol{\delta})$). We compare these alternatives to the scoring method used in Speculative RAG ($\rho = \texttt{Score}(\alpha|Q, \beta)$) in Table 2. The results show that incorporating longer retrieved documents does not consistently improve performance and tends to increase latency. This suggest that the generated rationale is already of high quality and serves as an effective bridge between the supporting documents and the generated answer drafts. By leveraging this rationale, we can efficiently verify drafts using a generic LM, leading to accurate final results. We further validate the rationale generation in the instruction-tuning stage. See Appendix D for more details.

## 4.5 Latency Analysis with Baselines

We analyze the latency of Standard RAG, Self-RAG, and our Speculative RAG on TriviaQA, MuSiQue, PopQA, PubHealth, and ARC-Challenge. We randomly sample 100 cases from each dataset and report the average time cost for each case, as shown in Figure 3. To simulate real-world application scenarios, we process cases individually without batching. As representative example, we run $\mathcal{M}_{\text{Verifier-8x7B}} + \mathcal{M}_{\text{Drafter-7B}}$ for Speculative RAG and Mixtral-Instruct$_{8x7B}$ for Standard RAG, as these demonstrate the highest performance among competitive baselines (see Table 1). We also include the analysis for Standard RAG: Mistral-Instruct$_{7B}$ and Self-RAG: Mistral-Instruct$_{7B}$ in this study. For Speculative RAG, we launch 5 endpoints of $\mathcal{M}_{\text{Drafter-7B}}$ for parallel drafting on TriviaQA, PopQA, PubHealth, and ARC-Challenge. We launch 10 endpoints for MuSiQue due to more drafts. We use tensor parallelism of 4 to fit Mixtral-Instruct$_{8x7B}$ into the GPU memory. We use the same tensor parallelism setting for the other methods for a fair comparison.

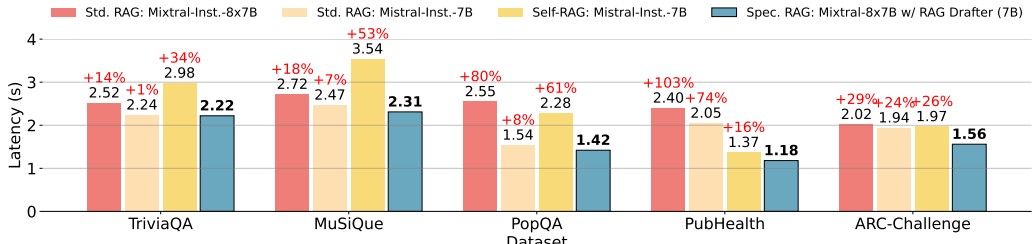

Figure 3: Latency analysis of Standard RAG, Self-RAG, and SPECULATIVE RAG on TriviaQA, MuSiQue, PopQA, PubHealth, and ARC-Challenge. The latency difference between Standard RAG/Self-RAG and SPECULATIVE RAG is highlighted in *red* $(+x\%)$. The latency varies across different datasets due to different retrieved document lengths. SPECULATIVE RAG encodes the retrieved documents in parallel and generates answer drafts with a smaller RAG drafter. This significantly improves the efficiency.

**Reducing processing time while maintaining high performance**  As the results demonstrate, SPECULATIVE RAG consistently achieves the lowest latency compared to all other methods. This advantage comes from its utilization of fewer documents needed per draft and parallel drafting. Particularly, compared with the most competitive baseline, Standard RAG: $\mathcal{M}_{\text{Verifier-8x7B}} + \mathcal{M}_{\text{Drafter-7B}}$, our proposed SPECULATIVE RAG reduces latency by up to 11.90% on TriviaQA, 15.07% on MuSiQue, 44.31% on PopQA, 50.83% on PubHealth, and 22.77% on ARC-Challenge. Furthermore, a direct comparison between Standard RAG: Mistral-Instruct$_{7B}$ and our method reveals that the higher latency of Standard RAG: Mistral-Instruct$_{7B}$ is due to its longer context length which contains all retrieved documents. Self-RAG: Mistral-Instruct$_{7B}$ also exhibits higher latency due to the generation of longer answers with self-reflection tags and the additional overhead associated with evidence selection. These findings highlight the advantage of our approach in reducing processing time while maintaining high performance.

## 4.6   ABLATION STUDIES

We conduct ablation studies on the multi-perspective sampling (Section 3.2) and the evaluation scores (Section 3.3) of SPECULATIVE RAG during the drafting or the verification stages on TriviaQA and PubHealth in Table 3. We use $\mathcal{M}_{\text{Verifier-8x7B}} + \mathcal{M}_{\text{Drafter-7B}}$ as a running configuration. Same as the main results, we report the accuracy as performance metrics.

**Diversity and reduced redundancy in retrieval improves draft quality significantly.**  In the first set of experiments, we evaluate the impact of multi-perspective sampling during the drafting. Recall that SPECULATIVE RAG clusters retrieved documents into distinct perspectives and sample one document from each cluster to reduce redundancy for the draft generation. We compare this against two alternative sampling strategies: (1) Random sampling without multi-perspective clustering, where we randomly select a document subset as context, and (2) Always sampling from the same cluster, where we select all documents from a single cluster. Our results indicate that our proposed sampling method yields the best performance thanks to its ability to leverage diverse context. Particularly, it improves the accuracy up to 1.88% on TriviaQA and 2.23% on PubHealth. While random sampling without clustering introduces diversity, it is prone to including redundant documents, degrading draft quality. Sampling from the same cluster significantly underperforms due to a lack of diverse perspectives.

**Scoring method on self-consistency and self-reflection refines draft quality effectively.**  In the second set of experiments, we examine the scoring method during verification. We remove each of the specific confidence scores, $\rho_{\text{Draft}}$, $\rho_{\text{Self-contain}}$, or $\rho_{\text{Self-reflect}}$ in turn. Performance drops are observed when any score is removed. Particularly, removing $\rho_{\text{Draft}}$ leads to a minimal decline, 0.19% on TriviaQA and 1.12% on PubHealth, likely due to the limited verification capability of the smaller RAG drafter. Removing either $\rho_{\text{Self-contain}}$ or $\rho_{\text{Self-reflect}}$ results in similar performance decreases, around 2.0% on TriviaQA and around 0.8% on PubHealth, indicating that both self-containment and self-reflection capture different key aspects of reasoning and are crucial during verification. Random selection without verification leads to substantial underperformance, resulting in a performance decline of 5.69% on TriviaQA and 5.37% on PubHealth.

Table 3: Ablation study of SPECULATIVE RAG in the drafting and verification stages on TriviaQA and PubHealth.

| | TriviaQA | PubHealth |
|---|---|---|
| $\mathcal{M}_{\text{Verifier-8x7B}} + \mathcal{M}_{\text{Drafter-7B}}$ | **74.24** | **76.60** |
| *Drafting Stage* | | |
| Random sampling *w/o* multi-perspective clustering | 73.02 (-1.22) | 75.38 (-1.22) |
| Always sampling from the same perspective cluster | 72.36 (-1.88) | 74.37 (-2.23) |
| *Verification Stage* | | |
| *w/o* $\rho_{\text{Draft}}$ ($\rho = \rho_{\text{Self-contain}} \cdot \rho_{\text{Self-reflect}}$) | 74.05 (-0.19) | 75.48 (-1.12) |
| *w/o* $\rho_{\text{Self-contain}}$ ($\rho = \rho_{\text{Draft}} \cdot \rho_{\text{Self-reflect}}$) | 72.04 (-2.20) | 75.89 (-0.71) |
| *w/o* $\rho_{\text{Self-reflect}}$ ($\rho = \rho_{\text{Draft}} \cdot \rho_{\text{Self-contain}}$) | 72.36 (-1.88) | 75.68 (-0.92) |
| Random selection *w/o* verification | 68.55 (-5.69) | 71.23 (-5.37) |

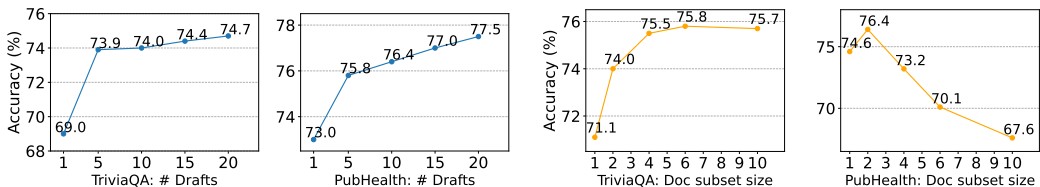

(a) We include 1, 5, 10, 15, 20 drafts and sample 2 supporting documents for each draft.

(b) We sample 1, 2, 4, 6, 10 supporting documents for each draft and we generate 10 answer drafts.

Figure 4: Performance analysis of SPECULATIVE RAG with (a) different numbers of drafts, and (b) different supporting document subset size on TriviaQA and PubHealth.

### 4.7 EFFECTS OF DRAFT NUMBER AND DOCUMENT SUBSET SIZE

**Increasing the number of drafts improves performance without adding latency.** We investigate the performance of SPECULATIVE RAG under varying numbers of drafts. Using $\mathcal{M}_{\text{Verifier-7B}}$ + $\mathcal{M}_{\text{Drafter-7B}}$ with 1, 5, 10, 15, 20 drafts on TriviaQA and PubHealth. We sample two documents as context per draft. The results are illustrated in Figure 4(a). Since we retrieve top 10 documents in total, we sample up to 20 drafts in these experiments. The results indicate that incorporating more drafts can further improve performance, likely thanks to higher coverage of diverse perspective of documents. Importantly, in SPECULATIVE RAG, we can launch multiple RAG drafter instances to generate drafts in parallel without additional latency.

**Increasing the document subset size doesn't always lead to better performance.** We also examine the effect of document subset size. By varying the number of documents (1, 2, 4, 6, or 10) sampled for draft generation on TriviaQA and PubHealth (Figure 4(b)), we find that including more documents in the context does not always lead to consistent performance improvement. While TriviaQA queries may benefit from more supporting documents due to their complexity, $\mathcal{M}_{\text{Verifier-7B}}$ + $\mathcal{M}_{\text{Drafter-7B}}$ can surpass Mistral-Instruct$_{7B}$ even with a single supporting document per draft. Furthermore, with two or more documents per draft, $\mathcal{M}_{\text{Verifier-7B}}$ + $\mathcal{M}_{\text{Drafter-7B}}$ can even surpass Mixtral-Instruct$_{8x7B}$. This further demonstrates the effectiveness of our drafting design.

## 5 CONCLUSION

Our proposed SPECULATIVE RAG decomposes RAG tasks into two separate steps of drafting followed by verification. SPECULATIVE RAG delegates the heavy lifting of drafting to a small specialized RAG drafter, while verification is done using a large generalist LM. The parallel generation of multiple drafts from diverse document subsets provides high quality answer candidates while reducing input token counts and the potential risk of position-bias-over-long-context, resulting in substantial improvements in both the quality and speed of the final output generation. We demonstrate the effectiveness of SPECULATIVE RAG with accuracy gains up to 12.97% while reducing latency by 50.83% compared to conventional RAG systems. SPECULATIVE RAG sheds new light on the potential of collaborative architectures for enhancing RAG performance through task decomposition.

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

APPENDIX

## A  INSTRUCTION-TUNING SETTINGS

We construct our training dataset for the RAG drafter from diverse instruction-following pairs. We sample instances from Open-Instruct processed data (Wang et al., 2023a) and knowledge-intensive datasets (Petroni et al., 2021; Stelmakh et al., 2022; Mihaylov et al., 2018). We augment the instruction-following pairs with retrieved documents and generated rationale. We use the off-the-shelf dense retriever Contriever-MS MARCO (Izacard et al., 2021) to retrieve up to 10 documents for each pair and use Gemini-Ultra (Team et al., 2023) to generate rationale. In total, we acquire a dataset of 40k instances. We use Mistral$_{7B}$ (v0.1) as our base LM for the RAG drafter. We reproduce the performance of Self-RAG (Asai et al., 2023) and CRAG (Yan et al., 2024) with Mistral$_{7B}$ (v0.1) for a fair comparison. We implement the training scripts using the Transformers library from Hugging Face (Wolf et al., 2019). We employ DeepSpeed (Rasley et al., 2020) to accelerate the training process. All experiments are conducted on a Linux server equipped with 16 Nvidia A100-SXM4-40GB GPUs.

Additionally, we replace Gemini-Ultra (Team et al., 2023) with GPT-4o (Achiam et al., 2023) when curating the instruction-tuning data for our RAG drafter to investigate the effects of different LLMs. These results demonstrate that SPECULATIVE RAG maintains its performance advantage, even when trained on data curated by GPT-4o. It consistently outperforms the baselines from Standard RAG, SelfRAG, and CRAG, further validating the effectiveness of our approach.

Table 4: RAG results on TriviaQA, PubHealth, ARC-Challenge with the RAG drafter trained on instruction-tuning data curated by GPT-4o and Gemini-Ultra.

|  | Trivial QA | PubHealth | ARC-C |
|---|---|---|---|
| $\mathcal{M}_{\text{Verifier-7B}} + \mathcal{M}_{\text{Drafter-7B}}^{\text{Gemini-U}}$ | 73.91 | 75.79 | 76.19 |
| $\mathcal{M}_{\text{Verifier-8x7B}} + \mathcal{M}_{\text{Drafter-7B}}^{\text{Gemini-U}}$ | 74.24 | 76.60 | 80.55 |
| $\mathcal{M}_{\text{Verifier-7B}} + \mathcal{M}_{\text{Drafter-7B}}^{\text{GPT-4o}}$ | 72.24 | 73.05 | 76.54 |
| $\mathcal{M}_{\text{Verifier-8x7B}} + \mathcal{M}_{\text{Drafter-7B}}^{\text{GPT-4o}}$ | 73.58 | 73.35 | 80.63 |

Moreover, we use an instruction-tuned Gemma-2$_{2B}$ (Team et al., 2024) as the RAG drafter and the frozen Mistral$_{7B}$ or Mixtral$_{8x7B}$ as the RAG verifier. We report the performance analysis in Table 5. These results suggest that Gemma-2$_{2B}$ provides a promising avenue for future work of further optimization.

Table 5: RAG results on TriviaQA, PubHealth, ARC-Challenge with an instruction-tuned Gemma-2$_{2B}$ or Mistral$_{7B}$ as the RAG drafter and a Mixtral$_{8x7B}$ as the RAG verifier.

|  | Trivial QA | PubHealth | ARC-C |
|---|---|---|---|
| $\mathcal{M}_{\text{Verifier-8x7B}} + \mathcal{M}_{\text{Drafter-2B}}$ | 67.22 | 72.14 | 67.92 |
| $\mathcal{M}_{\text{Verifier-8x7B}} + \mathcal{M}_{\text{Drafter-7B}}$ | 74.24 | 76.60 | 80.55 |

## B  EFFECTS OF INSTRUCTION TUNING

In SPECULATIVE RAG, we introduce a framework that combines the RAG drafter and the verifier. In this ablation study, we directly feed all documents to the RAG drafter and generate one draft ($m = 1$, $k = $ total # of documents). As shown in Table 6, we observe that instruction tuning effectively enhances the document understanding capability of the RAG drafter, as it outperforms both Mistral$_{7B}$ and Mistral-Instruct$_{7B}$. However, there remains a gap compared to SPECULATIVE RAG, showing the effectiveness of the drafting and verification framework.

Table 6: RAG results on TriviaQA and PubHealth ($m = 1$, $k$ = total # of docs)

| | TrivialQA | PubHealth |
|---|---|---|
| Mistral$_{7B}$ | 54.15 | 34.85 |
| Mistral-Instruct$_{7B}$ | 67.11 | 42.15 |
| $\mathcal{M}_{\text{Drafter-7B}}$ ($m = 1$, $k = 10$) | 73.23 | 65.25 |
| $\mathcal{M}_{\text{Verifier-7B}} + \mathcal{M}_{\text{Drafter-7B}}$ ($m = 5$, $k = 2$) | **73.91** | **75.79** |

## C  EFFECTS OF SELF-REFLECTION STATEMENT

We use "Do you think the explanation supports the answers? (Yes or No)" as the self-reflection statement in our main results. In this study, we replace it with other alternatives to see how the self-reflection statement affects the accuracy. The results are reported in Table 7. We observe that the performance does not change a lot given different self-reflection statements, which shows the stable verification capability of the generalist LMs by language modeling objective.

Table 7: Performance analysis of SPECULATIVE RAG with different self-reflection statements $R$ when computing the self-reflection score $\rho_{\text{Self-reflect}} = P(\texttt{"Yes"}|Q, \alpha, \beta, R)$, where $Q$ is the query, $\alpha, \beta$ are the generated answer draft and rationale.

| Reflection Statement | TriviaQA | PubHealth |
|---|---|---|
| *Do you think the explanation supports the answers? (Yes or No)* | 74.24 | 76.60 |
| *Does the rationale support the answer? (Yes or No)* | 74.22 | 76.09 |
| *What do you think about the rationale? A good one? (Yes or No)* | 74.25 | 75.79 |
| *Is the rationale good enough to support the answer? (Yes or No)* | 74.39 | 76.29 |

## D  EFFECTS OF RATIONALE GENERATION

We acknowledge that the generation of rationale potentially increases the inference cost during the drafting stage while this is crucial for the verifier in our method to assess the quality and reliability of generated drafts. And, the potential overhead can be mitigated through efficient parallel inference.

To further study the impact of rationale generation, we finetune the RAG drafter without rationale. We denote this setting as: *without rationale in drafting*. Similarly, *with rationale/doc in verification* indicates that we use the generated rationale or the retrieved documents as reference during the verification stage. We use $\mathcal{M}_{\text{Verifier-8x7B}} + \mathcal{M}_{\text{Drafter-7B}}$ as a running example. The results are shown in Table 8.

Table 8: Ablation study on the draft generation in the drafting and verification stages on TriviaQA and PubHealth.

| Drafting | Verification | TrivialQA | PubHealth | ARC-C |
|---|---|---|---|---|
| *w/o rationale* | *w/ doc* | 70.86 | 75.28 | 78.67 |
| *w/ rationale* | *w/ doc* | 74.08 | 76.09 | 80.46 |
| *w/ rationale* | *w/ rationale* | 74.24 | 76.60 | 80.55 |

**Better answer drafting**  As explored in Hsieh et al. (2023), incorporating rationale generation during instruction-tuning can lead to the RAG drafter producing higher-quality answer drafts. The results in Table 8 clearly demonstrate this. We observe a significant performance drop across all three benchmarks when the RAG drafter is finetuned without the rationale component,

**Lower latency and cost in verification**  We verify each draft against the rationale instead of the retrieved documents. From the ablation results, these generated rationales serve as high-quality grounding facts, improving verification performance compared to using the retrieved documents.

# E  EFFECTS OF DIFFERENT VOLUME OF TRAINING DATA

We acknowledge the importance of evaluating our framework's performance across different training data volumes. Our primary experiment utilized 40,059 instances to train our drafter model. To thoroughly assess scaling effects, we conducted additional experiments using incremental subsets of 10,000, 20,000, and 30,000 training instances. The results of these systematic evaluations are detailed in Table 9.

Table 9: Performance analysis of SPECULATIVE RAG with different volumn of instruction-tuning data.

| $\mathcal{M}_{\text{Verifier-8x7B}} + \mathcal{M}_{\text{Drafter-7B}}$ | TriviaQA | PubHealth |
|---|---|---|
| 10,000 | 71.69 | 72.34 |
| 20,000 | 72.64 | 72.44 |
| 30,000 | 73.20 | 74.37 |
| **Total (40,059)** | **74.24** | **76.60** |

From the table, we can conclude that increasing the volume of training data leads to improved performance. Specifically, the model's accuracy continues to rise as more instances are included, with the highest performance observed at 40,059 instances. This suggests that larger training datasets contribute positively to the performance of our drafter-verifier framework, indicating that scaling up data size could enhance the robustness of the model.

# F  EFFICACY OF SPECULATIVE RAG IN MULTI-HOP REASONING

We further validate SPECULATIVE RAG in the scenario of multi-hop reasoning. One of the key challenges of multi-hop reasoning is to effectively combine multiple pieces of evidence to arrive at the final answer. Indeed, the ability to verify or contrast information across documents is crucial to solve complex questions. We compare the performance of SPECULATIVE RAG with baselines on MuSiQue (Trivedi et al., 2022) and HotpotQA (Yang et al., 2018), two multi-hop reasoning benchmarks. We randomly sample 500 examples from the validation set of HotpotQA as the test set in our experiment. We adopt the same setting as MuSiQue on HotpotQA. The results are in Table 10. We find that our SPECULATIVE RAG achieves the best performance. Specifically, SPECULATIVE RAG improves accuracy by 2.15% on MuSiQue and by a substantial 5.4% on HotpotQA.

Table 10: RAG results on MuSiQue and HotpotQA

| | MuSiQue | HotpotQA |
|---|---|---|
| Mixtral-Instruct$_{\text{8x7B}}$ | 29.42 | 43.60 |
| Self-RAG$_{\text{Mistral-7B}}$ | 21.72 | 27.20 |
| $\mathcal{M}_{\text{Verifier-7B}} + \mathcal{M}_{\text{Drafter-7B}}$ | 31.03 | 47.60 |
| $\mathcal{M}_{\text{Verifier-8x7B}} + \mathcal{M}_{\text{Drafter-7B}}$ | **31.57** | **49.00** |

Our approach tackles this challenge by multi-perspective sampling when selecting documents for each draft (Section 3.2). We cluster the retrieved documents into distinct topics using an instruction-aware embedding model (Peng et al., 2024). Then, we sample one document from each cluster to form a diverse document subset, ensuring each drafter receives a variety of perspectives from the retrieval results. To validate the efficacy of this strategy, we further conduct an ablation study on MuSiQue and HotpotQA in Table 11. From the table, our sampling strategy effectively guarantees the diversity of information within the supporting document subsets, leading to improved performance of SPECULATIVE RAG on these tasks.

## F.1  PERFORMANCE BREAKDOWN ON HOTPOTQA

HotpotQA includes two types of quesitions: bridge-type questions in HotpotQA require a two-step reasoning process where the answer to the first step is crucial for answering the second. For example:

Table 11: Ablation study of multi-perspective sampling on multi-hop reasoning benchmarks: MuSiQue, HotpotQA.

|  | **MuSiQue** | **HotpotQA** |
|---|---|---|
| Random sampling | 29.33 | 48.2 |
| Multi-perspective sampling | **31.57** | **49.00** |

- *"When was the singer and songwriter of Radiohead born?"*
    - Step 1: Who is the singer and songwriter of Radiohead? $\rightarrow$ Thom Yorke
    - Step 2: When was [Thom Yorke](answer of step 1) born? $\rightarrow$ October 7, 1968
    - Final answer: October 7, 1968

In contrast, comparison-type questions also involve two steps, but the answers to each step are independent of each other. For example:

- *"Who was born first, Morgan Llywelyn or Robert Jordan?"*
    - Step 1: What's Morgan Llywelyn's DOB? $\rightarrow$ December 3, 1937
    - Step 2: What's Robert Jordan's DOB? $\rightarrow$ October 17, 1948
    - Final answer: Morgan Llywelyn

Table 12: Performance of SPECULATIVE RAG for different question types

| **Question Type** | **# of Questions** | SPECULATIVE RAG |
|---|---|---|
| Bridge-type | 400 | 41.75 |
| Comparison-type | 100 | 78.00 |
| Overall | 500 | 49.00 |

We report the performance breakdown of SPECULATIVE RAG on HotpotQA in Table 12. The results demonstrate a superior performance on comparison-type questions with multi-perspective sampling. This aligns with our expectations, as multi-perspective sampling ensures the document subset covers the diverse topics necessary for answering comparison-type questions. Revisiting the example above, *"Who was born first, Morgan Llywelyn or Robert Jordan?"*, with $k = 4$, our approach clusters retrieved documents into four groups. Group 0 and 3 focus on Morgan, while group 1 and 2 focus on Robert. As we sample one document from each group for the drafters, this clustering result ensures each drafter receives documents about both individuals. This balanced information distribution is crucial for the comparison-type questions. In contrast, random sampling risks providing a drafter with information about only one person, yielding a suboptimal draft.

On the other hand, we also observe that the multi-perspective sampling is less helpful for bridge-type questions. These questions require the LLM to first identify the *"bridge entity"* (e.g., Thom Yorke in the earlier example), a task our current work isn't explicitly designed for. While multi-perspective sampling effectively covers different topics in the drafts and the map-reduce approach accelerates inference, they might not directly contribute to pinpointing the "bridge entity" - the key to answering bridge-type questions.

We believe our framework could be effectively combined with other techniques specifically designed for bridge-type questions, such as those proposed in Xia et al. (2024b); Jiapeng et al. (2024); Shi et al. (2024). For instance, the Tree-of-Reviews (ToR) framework, introduced in Jiapeng et al. (2024), addresses multi-hop reasoning problems by dynamically initiating new searches based on previously retrieved documents and constructing various reasoning paths. This dynamic searching strategy can be integrated into our SPECULATIVE RAG, enabling each drafter to answer bridge-type questions more effectively.

## G PROMPT OF RATIONALE GENERATION

```
======================================= Prompt =======================================

# Memorize this piece of evidence in mind and use it as if you already know it.
# Evidence: State religion
Despite enjoying considerable popular support, Christianity was still not the official
state religion in Rome, although it was in some neighboring states such as Armenia, Iberia,
 and Aksum. Roman Religion (Neoplatonic Hellenism) was restored for a time by the Emperor
Julian from 361 to 363. Julian does not appear to have reinstated the persecutions of the
earlier Roman emperors. Catholic Christianity, as opposed to Arianism and other ideologies
deemed heretical, was declared to be the state religion of the Roman Empire on 27 February
380 by the decree "De fide catolica".

# Instruction: When did christianity become official religion of Rome?

# Response: 380 AD

# Rationale: (complete the following sentence with details from the evidence; you can only
use the information from the evidence)

===================================== Completion =====================================

Catholic Christianity was declared the official state religion of the Roman Empire on
February 27th, 380, through the decree "De fide catolica".
```

Figure 5: Prompt of Rationale Generation for Gemini-Ultra

## H PROMPT OF RAG DRAFTING

```
======================================= Prompt =======================================

Response to the instruction. Also provide rationale for your response.
## Instruction: In Buddhism, what is the state of blissful repose or absolute existence by
someone relieved of the necessity of rebirth?
## Evidence:
[1] Buddhism
Nirvana literally means "blowing out, quenching, becoming extinguished". In early Buddhist
texts, it is the state of restraint and self-control that leads to the "blowing out" and
the ending of the cycles of sufferings associated with rebirths and redeaths. Many later
Buddhist texts describe nirvana as identical with "anatta" with complete "emptiness,
nothingness". In some texts, the state is described with greater detail, such as passing
through the gate of emptiness ("sunyata") realizing that there"
[2] Salvation
It includes a variety of disciplines, such as yoga and meditation. Nirvana is the profound
peace of mind that is acquired with moksha (liberation). In Buddhism and Jainism, it is the
 state of being free from suffering. In Hindu philosophy, it is union with the Brahman (
Supreme Being). The word literally means "blown out" (as in a candle) and refers, in the
Buddhist context, to the blowing out of the fires of desire, aversion, and delusion, and
the imperturbable stillness of mind acquired thereafter. In Theravada Buddhism the emphasis
 is on one's

===================================== Completion =====================================
## Rationale: Nirvana literally means 'blowing out, quenching, becoming extinguished'. It
is described as a state of "restraint and self-control" that leads to the "blowing out" and
 the ending of the cycles of sufferings associated with rebirths and redeaths.
## Response: In Buddhism, the state of blissful repose or absolute existence by someone
relieved of the necessity of rebirth is called Nirvana.
```

Figure 6: Prompt of RAG Drafting

## I PROMPT OF STANDARD RAG

```
====================================== Prompt ======================================

Below is an instruction that describes a task. Write a response that appropriately
completes the request.

### Evidence:
[1] Britain (place name)
Britain, after which "Britain" became the more commonplace name for the island called Great
 Britain. After the Anglo-Saxon period, "Britain" was used as a historical term only.
Geoffrey of Monmouth in his pseudohistorical "Historia Regum Britanniae" ...

[2] Great Britain
The peoples of these islands of "Prettanike" were called the "Priteni" or "Pretani". "
Priteni" is the source of the Welsh language term Prydain, "Britain", which has the same
source as the Goidelic term Cruithne used to refer to the early Brythonic-speaking
inhabitants of Ireland. The latter were later called Picts or Caledonians ...

...

[10] Albion
Albion is an alternative name for Great Britain. The oldest attestation of the toponym
comes from the Greek language. It is sometimes used poetically and generally to refer to
the island, but is less common than 'Britain' today. The name for Scotland in most of the
Celtic languages is related to Albion: "Alba" in Scottish Gaelic, "Albain" ...

### Instruction: What was Britain called - before it was Britain?

### Response:
```

Figure 7: Prompt of Standard RAG for Non-instruction-tuned LM

```
====================================== Prompt ======================================

[INST] Below is an instruction that describes a task. Write a response for it and state
your explanation supporting your response.

### Instruction: What was Britain called - before it was Britain?
### Evidence:
[1] Britain (place name)
Britain, after which "Britain" became the more commonplace name for the island called Great
 Britain. After the Anglo-Saxon period, "Britain" was used as a historical term only.
Geoffrey of Monmouth in his pseudohistorical "Historia Regum Britanniae" ...

[2] Great Britain
The peoples of these islands of "Prettanike" were called the "Priteni" or "Pretani". "
Priteni" is the source of the Welsh language term Prydain, "Britain", which has the same
source as the Goidelic term Cruithne used to refer to the early Brythonic-speaking
inhabitants of Ireland. The latter were later called Picts or Caledonians ...

...

[10] Albion
Albion is an alternative name for Great Britain. The oldest attestation of the toponym
comes from the Greek language. It is sometimes used poetically and generally to refer to
the island, but is less common than 'Britain' today. The name for Scotland in most of the
Celtic languages is related to Albion: "Alba" in Scottish Gaelic, "Albain" ...
[/INST] The response is:
```

Figure 8: Prompt of Standard RAG for Instruction-tuned LM

## J    CASE STUDY

Figure 9 and 10 are two representative cases from TriviaQA and PubHealth. They show the two drafts generated for the same question. We observe that our RAG drafter can well understand the multiple perspectives in the retrieval results and generate high-quality drafts. Our RAG verifier can also help filter out the unreliable drafts as we observe a relatively low scores in the first draft in Figure 9 and 10.

```
================================= Prompt of Drafter ==================================

Response to the instruction. Also provide rationale for your response.
## Instruction: Which actress/singer starred as Doralee Rhodes in the 1980 film, "Nine to
Five"?
## Evidence:
[1] Diana DeGarmo
Sheila until the show's closure on June 27, 2010. DeGarmo played the role of Doralee Rhodes
 in the national tour of "9 to 5", which was launched in Nashville on September 21, 2010.
She ended her run as Doralee after the July 2011 Minneapolis tour stop. DeGarmo appeared on
 musical theatre composer Scott Alan's third album, ...
[2] 9 to 5 (musical)
Pippa Winslow as Violet Newstead, Louise Olley as Doralee Rhodes and Leo Sene as Franklin
Hart Jr, with Samantha Giffard as Roz, Matthew Chase as Joe and Mark Houston, Rachel Ivy,
and Blair Anderson. "9 to 5" will play in the West End at the Savoy Theatre from January 29
 to August 31, 2019. The production stars Amber Davies (Judy), ...

================================= Completion =====================================
## Rationale: Diana DeGarmo played the role of Doralee Rhodes in the national tour of "9 to
 5", which began in September 2010.
## Response: Diana DeGarmo

================================= Verification =====================================
draft score: 0.6594
self-containment score: 0.3417
self-reflection score: 0.5238

------------------------------------------------------------------------------------

================================= Prompt of Drafter ==================================

Response to the instruction. Also provide rationale for your response.
## Instruction: Which actress/singer starred as Doralee Rhodes in the 1980 film, "Nine to
Five"?
## Evidence:
[1] Cynthia Rhodes
Rhodes garnered her first non-dance related role in Michael Crichton's 1984 science fiction
 thriller "Runaway" with Tom Selleck, Kirstie Alley and Gene Simmons. Her most notable role
 was as dance instructor Penny Johnson in the hit 1987 motion picture "Dirty Dancing" with
Jennifer Grey and Patrick Swayze. Rhodes' final motion picture role was the character of
Vickie Phillips opposite actor Jameson Parker in the sleeper action-adventure movie ...
[2] Dolly Parton
Parton's feature film "9 to 5" (1980). The musical ran at the Ahmanson Theatre, Los Angeles
 in late 2008. It opened on Broadway at the Marquis Theatre in New York City, on April 30,
2009, to mixed reviews. The title track of her 2008 "Backwoods Barbie" album was written
for the musical's character Doralee. Although her score (as well as the musical debut of
actress Allison Janney) were praised, ...

================================= Completion =====================================
## Rationale: Dolly Parton starred in the 1980 film "9 to 5" as Doralee Rhodes.
Additionally, Parton's 2008 musical "9 to 5" was based on the film, with Parton reprising
her role as Doralee.
## Response: Dolly Parton starred as Doralee Rhodes in the 1980 film, "Nine to Five".

================================= Verification =====================================
draft score: 0.71
self-containment score: 0.4346
self-reflection score: 0.7449
```

Figure 9: Case study of SPECULATIVE RAG from TriviaQA where *Dolly Parton* is the correct answer.

```
================================ Prompt of Drafter ===================================

Response to the instruction. Also provide rationale for your response.
## Instruction: A lot of the problems with forest fires is because of bad policy (not to
clear out the forests) that was put in place by environmentalists.
## Evidence:
[1] Forest restoration
That can be because of provoked deforestation in order to substitute forests by crop areas,
 or in dry areas, because of wild fires occurring naturally or intentionally. A whole
section of forest landscape restoration in linked to this particular problem, as in many
cases, the net loss of ecosystem value is very high and can open the drop to an accelerated
 further degradation of the soil conditions through erosion and desertification. This
indeed has dire consequences on both the quality of the habitats and their related fauna.
...
[2] Wildfire
Attempts to thin out the fuel load may be met with opposition due to the desirability of
forested areas. Wildland goals may be further resisted because of endangered species
protections and habitat preservation. The ecological benefit of fire is often overridden by
 the economic benefits of protecting structures and lives. Additionally, federal policies
that cover wildland areas usually differ from local and state policies that govern urban
lands. ...

================================ Completion ===================================
## Rationale: Inadequate forest management can contribute to the severity of wildfires.
## Response: True

================================ Verification ===================================
draft score: 0.8161
self-containment score:0.1360
self-reflection score: 0.8877

------------------------------------------------------------------------------------

================================ Prompt of Drafter ===================================

Response to the instruction. Also provide rationale for your response.
## Instruction: A lot of the problems with forest fires is because of bad policy (not to
clear out the forests) that was put in place by environmentalists.
## Evidence:
[1] 1997 Indonesian forest fires
The 1997 Indonesian forest fires were caused by changing land use which made the tropical
forest vulnerable to fire during a drought associated with that year's El Nino. Indonesian
forests have historically been resistant to burning even during long dry seasons and
despite the use of fire to clear land for swidden agriculture. The land use changes that
led to the fires were a combination of industrial-scale logging, draining peatlands for
conversion to oil palm and fast-growing tree plantations, and a massive government program
to drain swamps and convert them to rice paddies. ...
[2] Wildfire
Attempts to thin out the fuel load may be met with opposition due to the desirability of
forested areas. Wildland goals may be further resisted because of endangered species
protections and habitat preservation. The ecological benefit of fire is often overridden by
 the economic benefits of protecting structures and lives. Additionally, federal policies
that cover wildland areas usually differ from local and state policies that govern urban
lands. ...

================================ Completion ===================================
## Rationale: The 1997 Indonesian forest fires were caused by **land use changes**,
including industrial-scale logging, draining peatlands.
## Response: False

================================ Verification ===================================
draft score: 0.8288
self-containment score: 0.1325
self-reflection score: 0.9533
```

Figure 10: Case study of SPECULATIVE RAG from PubHealth where *False* is the correct answer.

