# OpenReview forum: "Speculative RAG: Enhancing Retrieval Augmented Generation through Drafting"
_ICLR.cc/2025/Conference — ICLR 2025 Poster_

### Official Review · Reviewer_dkh1 · 2024-10-30

**Soundness:** 3
**Presentation:** 3
**Contribution:** 3
**Rating:** 6
**Confidence:** 4

**Summary:**

This paper leverages the idea of speculative decoding to design a speculative RAG framework, which offloads computational burden to a smaller, specialist LM that serves as the RAG drafter for generalist LMs. Extensive experiments and ablation studies demonstrate the effectiveness of the proposed method.

**Strengths:**

1. The designed specialist LM is lightweight and allows to analyze multiple documents and produce RAG drafts in parallel, which effectively reduces the computation cost and increases the inference efficiency.
2. The distinction between specialist and generalist LMs clarifies their functional roles, reducing the risk of generalization degradation that may result from supervised fine-tuning (SFT) of a generalist LM. Specialists, optimized through targeted fine-tuning, focus on improving draft generation capabilities, thereby enhancing result accuracy.
3. Applying a clustering algorithm, i.e. K-means, to pre-group retrieved documents and then uniformly sampling from each group, i.e., multi-perspective sampling, can mitigate analysis inaccuracies caused by incomplete information, enhancing robustness in practical applications.

**Weaknesses:**

1. The quality of the SFT data constructed by the author remains unclear due to insufficient analysis. Additionally, further ablation studies are needed to evaluate the drafter’s performance across varying data volumes. Acquiring substantial SFT data through a proprietary model is resource-intensive. Therefore, it is essential to investigate whether a smaller dataset could yield satisfactory results or if a large volume of SFT data is indeed required to achieve optimal outcomes.
2. P(Yes) or SR is commonly applied in hallucination detection. However, [1] highlights that LLMs without target SFT or aligned, while LLMs may differentiate between correct and incorrect results, the probability itself, i.e., P(Yes), is either uncalibrated or demonstrates poor calibration performance. Although Table 3 shows that incorporating SR enhances overall performance, there is insufficient evidence to validate the intrinsic effectiveness of P(Yes). This limitation weakens the robustness of the proposed method. Could the author also add an ablation study to independently assess the effectiveness of P(Yes)?
3. It seems that results for key baseline models, such as CRAG and Self-CRAG, are missing for the three free-form datasets. This omission is concerning, as it is standard research practice to replicate baseline results when they are not available from prior studies, especially when the authors are comparing their approach with only a few previous works. Could the authors also include the performance metrics of CRAG and Self-CRAG on the free-form datasets for a more comprehensive comparison?
4. According to the results in Table 1, Mixtral-Instruct-8x7B has already achieved high scores on TriviaQA and ARC-C (73.91% and 78.41%), and the improvement brought by Speculative RAG is limited (74.24% and 80.55%). These results may diminish the contribution of the proposed method, as the performance improvement brought by instruction tuning is more pronounced and stable (Mistral-7B vs. Mistral-Instruct-7B and Mixtral-8x7B vs. Mixtral-Instruct-8x7B). Could the authors provide more explanation and analysis to clarify this result?
5. Although the datasets used in this paper include a variety of question-answering formats, such as free-form (short-form and multi-hop) and closed-set, they are all based on wiki-style or specific domains. To more convincingly demonstrate the method's effectiveness, the authors should further extend their evaluation to more realistic and domain-diverse RAG datasets, like FreshQA and BRIGHT. Could the author also discuss the potential challenges in applying their method to these datasets and propose specific experiments to show the performance?

[1] Kadavath et al., Language Models (Mostly) Know What They Know, 2022.

**Questions:**

1. While the author has shown that the proposed method is effective, it is important to note that the generalist LM has not undergone fine-tuning. RAG serves to supply additional information to the LLM to mitigate knowledge gaps. However, if the generalist LM itself lacks relevant knowledge, can it reliably evaluate the rationale and drafts generated by the specialist LM, particularly when the specialist’s output is inaccurate or includes hallucinations? To better understand this limitation, could the author provide boundary cases or conduct an error analysis?

---

> ### Author Response · Authors · 2024-11-23
> **Response to Reviewer dkh1 (1/2)**
>
> We thank the reviewer dkh1 for the valuable feedback. We address individual questions carefully below.
>
> ---
> ## Different Volume of Training Data
>
> We acknowledge the importance of evaluating our framework's performance across different training data volumes. Our primary experiment utilized 40,059 instances to train our drafter model. To thoroughly assess scaling effects, we conducted additional experiments using incremental subsets of 10,000, 20,000, and 30,000 training instances. The results of these systematic evaluations are detailed below.
>
> | Speculative RAG: Drafter 7b + Verifier 8x7b | TQA   | PubHealth |
> | -- | -- | -- |
> | 10k                                          | 71.69 | 72.34     |
> | 20k                                          | 72.64 | 72.44     |
> | 30k                                          | 73.20 | 74.37     |
> | Total (40,059)                               | **74.24** | **76.60**     |
>
> From the table, we can conclude that increasing the volume of training data leads to improved performance. Specifically, the model's accuracy continues to rise as more instances are included, with the highest performance observed at 40,059 instances. This suggests that larger training datasets contribute positively to the performance of our drafter-verifier framework, indicating that scaling up data size could enhance the robustness of the model.
>
> ---
> ## Ablation study to assess the effectiveness of P(Yes)
>
> Thank you for pointing this out! We believe this ablation study will further clarify the design of Speculative RAG. Below are the results of our ablation study, which we will include in our future draft.
>
> | Speculative RAG: Drafter 7b + Verifier 8x7b | TQA   | PubHealth |
> | -------------------------------------------- | ----- | --------- |
> | random                                       | 68.55 | 71.23     |
> | $\rho_{\text{self-reflect}}$          | 70.98 | 73.05     |
> | $\rho_{\text{self-reflect}} \cdot \rho_{\text{self-contain}}$             | 74.05 | 75.48     |
> | $\rho_{\text{self-reflect}} \cdot \rho_{\text{self-contain}} \cdot \rho_{\text{draft}}$     | **74.24** | **76.60**     |
>
> In the table, we use various score combinations to verify the generated drafts. We find that P(Yes) is effective in identifying the best draft candidate (superior to the random choice), and its performance improves further when combined with other evaluation scores. As noted by the reviewer, we believe that the P(Yes) score could contribute even more significantly to the final performance if the verification model undergoes sufficient instruction tuning.
>
> ---
> ## Reproducing the results of CRAG and Self-CRAG
>
> We have thoroughly investigated the problem framework presented in [1] and successfully generated additional results for both CRAG and Self-CRAG implementations on the new benchmark datasets. We plan to incorporate these findings into the main results table of our paper.
>
> |                                              | TQA   | PopQA |
> | -------------------------------------------- | ----- | ----- |
> | CRAG                                         | 59.03 | 49.46 |
> | Self-CRAG                                    | 65.43 | 56.11 |
> | Speculative RAG: Drafter 7b + Verifier 7b | 73.91 | 56.75 |
> | Speculative RAG: Drafter 7b + Verifier 8x7b | 74.24 | 57.54 |

---

> > ### Author Response · Authors · 2024-11-23
> > **Response to Reviewer dkh1 (2/2)**
> >
> > ---
> > ## Clarify on the improvements
> >
> > We thank the reviewer for raising the important point about comparing Mixtral-Instruct-8x7B with our Speculative RAG system (Drafter-7B + Mixtral-8x7B). To provide clarity: our approach involves direct fine-tuning of the Mistral-7B base model to function as a drafter, while employing the frozen Mixtral-8x7B as a verifier. It's crucial to emphasize that we utilize these base models in their pre-instruction-tuned state, as released by the Mistral team.
> >
> > Our Speculative RAG system offers distinct advantages over Mixtral-Instruct-8x7B. First, it requires substantially less instruction-following training data. Second, it involves fine-tuning significantly fewer parameters compared to Mixtral-Instruct-8x7B, as demonstrated in the table below.
> >
> > |                    | Mixtral-Instruct-8x7B                                | Speculative RAG                   |
> > | ------------------ | ---------------------------------------------------- | --------------------------------- |
> > | Training data      | Huggingface or other in-house SFT or preference data | Part of the Open-Instruct Dataset |
> > | Size               | About 381,000 (estimated by the number used in OLMo)                  | 40,059                  |
> > | Training Parameter | 8x7B                                                 | 7B                                |
> >
> > Given the reduced training costs, superior performance, low latency, and higher accuracy, we believe our Speculative RAG approach offers a compelling advantage.
> >
> > ---
> > ## About the additional benchmarks
> >
> > Thank you for pointing this out. FreshQA assesses how well large language models handle current events and evolving information through factual question-answering. Its challenges stem from the lengthy, dynamic web documents it uses, which test both the parameter memory and long-context comprehension abilities of LLMs. On the other hand, BRIGHT is a text retrieval benchmark requiring intensive reasoning to identify relevant documents, rather than a question-answering task. While both are realistic and significant benchmarks in the RAG research community, they may not align well with the goals of our proposed RAG system.
> >
> > Our Speculative RAG framework builds on the problem settings of Self-RAG and CRAG, where each test sample includes a knowledge-intensive QA pair and multiple retrieved evidence pieces. The benchmarks we use—TriviaQA, PopQA, PubHealth, among others—span diverse domains and require precise retrieval to generate accurate answers. The main challenge lies in filtering irrelevant information and reasoning effectively over the additional retrieved data.
> >
> > In light of the above, we plan to incorporate these benchmark discussions in our next draft and explore adapting Speculative RAG to address the novel problem settings presented in these papers in our future study.
> >
> > ---
> > ## Error Case Study
> >
> > Thanks for pointing this out! We will include a case study on the error cases in our future draft. Here is an example.
> >
> > - **Question:** "Which US State ended prohibition in November 1948, a law that had been in place there for 68 years?"
> > - **Groundtruth:** State of Kansas
> > - **Retrieved Docs:** *(Most documents are about the background of the prohibition w/o mentioning Kansas. One document mentions Mississippi is the last state to repeal it.)*
> > - **Predicted Answer:** The state of Mississippi
> > - **Generated Rationale:** Mississippi was the last state to repeal its statewide prohibition laws in 1966.
> >
> > We review the answer drafts, their associated rationales, and the retrieved documents. Upon analysis, we find that none of the retrieved documents provide the correct answer, making it impossible to resolve the question accurately. Most documents discuss the background of prohibition without specifying a particular state. Only one document notes that *"By 1966, however, all states had repealed their statewide prohibition laws, with Mississippi being the last state to do so."* Our speculative RAG system was misled by this document, ultimately selecting the draft based on it as the final answer.

---

> > > ### Author Response · Authors · 2024-11-25
> > >
> > > Dear Reviewer dkh1, we're reaching out to see if you could take a look at our rebuttal ahead of the deadline. We value your perspective and would welcome your feedback.

---

> > > > ### Comment · Reviewer_dkh1 · 2024-11-29
> > > > **Thank you for the rebuttal**
> > > >
> > > > Thank you for the detailed responses. I acknowledge that I have read the responses and decided to keep my score.

---

> > > > > ### Author Response · Authors · 2024-12-03
> > > > >
> > > > > We sincerely thank Reviewer dkh1 for their valuable feedback and welcome any additional questions or feedback the reviewer may have.

---

### Official Review · Reviewer_niGd · 2024-11-01

**Soundness:** 3
**Presentation:** 3
**Contribution:** 2
**Rating:** 5
**Confidence:** 4

**Summary:**

This paper introduces Speculative RAG – a framework that leverages a larger generalist LM to efficiently verify multiple RAG drafts produced in parallel by a smaller, distilled specialist LM. Each draft is generated from a distinct subset of retrieved documents, offering diverse perspectives on the evidence while reducing input token counts per draft.

**Strengths:**

This paper trained an LLM (Drafter) to generate multiple answers paired with rationale (reference + explains). Then, they use another LLM (Verifier)to confirm the answers, which enhances the robustness.

**Weaknesses:**

The main contribution of this paper is insufficient. Because the whole framework just involves clustering the documents and fine-tuning small LLM to generate answers with rationale. Then prompt Verifier model confirms the answer based on rationale.

**Questions:**

Since this is a paper target for the RAG framework, testing the proposed framework on some RAG benchmarks may be more reliable. Because the QA benchmark is mainly designed for reasoning, not for the RAG framework.

---

> ### Author Response · Authors · 2024-11-23
> **Response to Reviewer niGd**
>
> We thank the reviewer niGd for the valuable feedback. We address individual questions carefully below.
>
> ---
> ## Contribution of Speculative RAG
>
> Inspired by Speculative Decoding [1], Speculative RAG is the first to apply the drafting paradigm to RAG problems. We share the core idea of using a smaller language model (LM) for efficient draft generation and a larger LM for verification and selection. As noted by reviewer GVM9, our approach is "both novel and appealing," and our empirical results demonstrate significant improvements in efficiency while maintaining high performance across benchmarks. We believe Speculative RAG has the potential to significantly impact the field of RAG by offering a practical solution for high-performance, low-latency retrieval-augmented generation.
>
> - [1] Leviathan, Yaniv, Matan Kalman, and Yossi Matias. "Fast inference from transformers via speculative decoding." International Conference on Machine Learning. PMLR, 2023.
>
> ---
> ## About the Benchmarks
>
> To evaluate Speculative RAG, we followed the standard RAG evaluation protocol established in prior work like Self-RAG [1], CRAG [2, 3], SAIL [4], InstructRAG [5, 6], and FlashRAG [7]. We used five well-known RAG benchmarks: TriviaQA, MuSiQue, PopQA, PubHealth, and ARC-Challenge. Each benchmark sample includes a knowledge-intensive question, its corresponding answer, and retrieved documents provided by the benchmark authors. We believe this comprehensive evaluation across diverse datasets provides strong evidence for the effectiveness of our method.
>
> - [1] Asai, Akari, et al. "Self-RAG: Learning to Retrieve, Generate, and Critique through Self-Reflection." The Twelfth International Conference on Learning Representations.
> - [2] Yan, Shi-Qi, et al. "Corrective retrieval augmented generation." arXiv preprint arXiv:2401.15884 (2024).
> - [3] Lewis, Patrick, et al. "Retrieval-augmented generation for knowledge-intensive nlp tasks." Advances in Neural Information Processing Systems 33 (2020): 9459-9474.
> - [4] Luo, Hongyin, et al. "Sail: Search-augmented instruction learning." arXiv preprint arXiv:2305.15225 (2023).
> - [5] Wei, Zhepei, Wei-Lin Chen, and Yu Meng. "InstructRAG: Instructing Retrieval-Augmented Generation with Explicit Denoising." arXiv preprint arXiv:2406.13629 (2024).
> - [6] Asai, Akari, et al. "Retrieval-based language models and applications." Proceedings of the 61st Annual Meeting of the Association for Computational Linguistics (Volume 6: Tutorial Abstracts). 2023.
> - [7] Jin, Jiajie, et al. "FlashRAG: A Modular Toolkit for Efficient Retrieval-Augmented Generation Research." arXiv preprint arXiv:2405.13576 (2024).

---

> > ### Comment · Reviewer_niGd · 2024-11-24
> >
> > Thank you for the comprehensive response to my review. However, I'm not confident that this paper should be strongly accepted, so I'll raise my score to 5.

---

> > > ### Author Response · Authors · 2024-11-25
> > >
> > > We thank Reviewer niGd for their careful consideration of our work and insightful comments. We appreciate their willingness to increase their score to 5, which we believe is a positive step towards acceptance. We are happy to address any further questions or concerns.

---

### Official Review · Reviewer_RYjP · 2024-11-04

**Soundness:** 3
**Presentation:** 3
**Contribution:** 3
**Rating:** 6
**Confidence:** 4

**Summary:**

This work introduces a new speculative decoding method to enhance the retrieval augmented generation. Specifically, a smaller distilled specialist LM is used to generate answer drafts based on randomly selected document subsets. After that, a larger generalist LM is used to verify the answer confidences according to the answer and rationale generation probabilities.

**Strengths:**

1. The paper is well-written and easy to follow. The authors present detailed descriptions of their methods, including prompts and experimental setups.
2. The proposed method achieves obvious improvements among the baselines, and the ablation studies can verify the effectiveness of their method.

**Weaknesses:**

1. The conducted experiments should include more recent RAG baselines, especially the speculative decoding methods (e.g., [1], [2]).
2. It is a trivial idea and makes limited contributions compared to previous studies. As mentioned in this paper, many recent studies investigate the two-stage RAG, where a drafting stage produces answer candidates and the assembling stage generates final answers based on the drafts. There are only a few differences between them.


[1] Ground every sentence: Improving retrieval-augmented llms with interleaved reference claim generation.

[2] RAT: Retrieval Augmented Thoughts Elicit Context-Aware Reasoning in Long-Horizon Generation.

**Questions:**

1. In Line 207-209, is the answer A obtained by human labeling or generated by LLMs given documents D? What if the D does not provide evidence for generating A?
2. Except for the inference latency, what are the consuming token numbers of different methods?

---

> ### Author Response · Authors · 2024-11-23
> **Response to Reviewer RYjP**
>
> We thank the reviewer RYjP for the valuable feedback. We address individual questions carefully below.
>
> ---
> ## More baseline methods
>
> Thank you for highlighting these relevant works! We will incorporate them into our final version.
>
> In [1], the authors introduce the ReClaim framework, which addresses knowledge-intensive questions with fine-grained, sentence-level citations. This enables models to provide detailed references for each sentence in question-answering tasks. In [2], the retrieval-augmented thoughts (RAT) framework is proposed, where each step of a chain of thought (CoT) is iteratively revised using retrieved information after generating an initial zero-shot CoT. These two related works focus on refining the rationale of generated answers, either by incorporating detailed citations or iteratively improving the reasoning chain. While these papers focus on refining the rationale for answering knowledge-intensive questions, Speculative RAG introduces a novel approach to efficiently generating multiple answer drafts in parallel using smaller LMs. These drafts are then rigorously verified by a larger LM, ensuring both efficiency and accuracy.
>
> We are particularly excited about the potential for integrating these approaches into future iterations of Speculative RAG. Combining the strengths of rationale refinement with our parallel generation and verification process could lead to even more robust and comprehensive answers to knowledge-intensive questions.
>
> - [1] Ground every sentence: Improving retrieval-augmented llms with interleaved reference claim generation.
> - [2] RAT: Retrieval Augmented Thoughts Elicit Context-Aware Reasoning in Long-Horizon Generation.
>
>
> ---
> ## Two-stage RAG
>
> While two-stage methods have been explored in various domains, our approach uniquely applies the draft-then-verify paradigm to RAG in a map-reduce fashion. This is a novel contribution that hasn't been thoroughly investigated in the context of RAG.
>
> Speculative RAG specifically addresses the need for a balance between efficiency and effectiveness in RAG. By employing a smaller, specialized LM to generate multiple drafts from distinct document subsets in parallel, we significantly reduce the computational burden. Subsequently, a larger, generalist LM verifies these drafts, selecting the most accurate one. This not only improves efficiency by offloading computation but also reduces input token counts for generation, leading to faster response times.
>
> Existing two-stage methods can be broadly categorized into those that generate a chain of thought before answering [1, 2, 3] and those that use a self-evolving process to refine answers iteratively [4, 5]. While sharing a two-stage structure, Speculative RAG's unique draft-and-verify design, inspired by Speculative Decoding, sets it apart and enables superior performance with reduced latency.
>
>
> - [1] Wang, Yile, et al. "Self-Knowledge Guided Retrieval Augmentation for Large Language Models." Findings of the Association for Computational Linguistics: EMNLP 2023. 2023.
> - [2] Wei, Zhepei, Wei-Lin Chen, and Yu Meng. "InstructRAG: Instructing Retrieval-Augmented Generation with Explicit Denoising." arXiv preprint arXiv:2406.13629 (2024).
> - [3] Wang, Zihao, et al. "Rat: Retrieval augmented thoughts elicit context-aware reasoning in long-horizon generation." arXiv preprint arXiv:2403.05313 (2024).
> - [4] Khattab, Omar, et al. "DSPy: Compiling Declarative Language Model Calls into State-of-the-Art Pipelines." The Twelfth International Conference on Learning Representations. 2024.
> - [5] Jiang, Zhengbao, et al. "Active Retrieval Augmented Generation." Proceedings of the 2023 Conference on Empirical Methods in Natural Language Processing. 2023.
>
> ---
> > **Q1:** in Line 207-209, is the answer A obtained by human labeling or generated by LLMs given documents D? What if the D does not provide evidence for generating A?
>
> The QA pairs are both from the open-source instruction-following datasets. In the cases where the retrieved documents are not relevant to the question or the answer cannot be derived from the documents, the LLM is instructed to generate a tag for us to filter this case out.

---

> > ### Author Response · Authors · 2024-11-25
> >
> > Dear Reviewer RYjP, we wanted to check if you might have a chance to review our rebuttal before the deadline. Your feedback would be greatly appreciated.

---

### Official Review · Reviewer_GVM9 · 2024-11-05

**Soundness:** 3
**Presentation:** 2
**Contribution:** 2
**Rating:** 5
**Confidence:** 4

**Summary:**

The paper proposed a retrieval-augmented generation framework termed SpeculativeRAG, leveraging high-level concepts analogical to speculative decoding.
The framework exhibits better performance with lower latency via lunching a set of smaller model instances in parallel, each processes a subset of retrieved documents, to produce answer drafts and rationales.
The answer drafts and rationales are subsequently verified by a larger, strong base LLM to select the final answer.

**Strengths:**

- The conceptual extension of the speculative farmwork proposed by the author is both novel and appealing, and supported by strong empirical results.

- The paper targets a timely challenge in RAG and offers promising techniques to improve the efficiency for the system.

- The experiments are comprehensive and thorough, in particular, the extensive ablation study and analysis provide great insights for a better understanding of the proposed approach.

**Weaknesses:**

- The instruction-tuning of the small drafter LMs requires synthesis rationales generated by Gemini Ultra from an additional 40k instances. This presents an unfair comparison for baselines methods that do not have access to these external resources in the experiments.

- It is unclear whether multiple instances of large verifier LLMs are also lunched in parallel. According to Line ```11~14``` in Algorithm ```1```, this seems to be the case. If so, the large memory overhead might significantly offset the latency gain in practical scenarios.

- A relevant work [1] (first released ~4.5 month ago) is missing in the baseline and not mentioned in the related work Section. To the best of my understanding, [1] also proposes a very similar synthesis rationale-augmented generation approach for RAG, thus, might be a suitable method for baseline comparison.

- The contribution and result of the paper could be further strengthened if newer generation of models (e.g., gemma-2, llama-3) are adopted in the experiments.

[1] Wei et al, *InstructRAG: Instructing Retrieval-Augmented Generation via Self-Synthesized Rationales*. 2024.

**Questions:**

- For the comparison of the average number of tokens in the generated rationale and the retrieved documents in Figure ```2```: What is the number of retrieved documents? Is it before or after subsetting to drafters?

- Is the Multi-Perspective Sampling stage also considered in the latency analysis in Section ```4.5```?

---

> ### Author Response · Authors · 2024-11-23
> **Response to Reviewer GVM9**
>
> We thank the reviewer GVM9 for the valuable feedback. We address individual questions carefully below.
>
> ---
> ## Access to these external resources
>
> In our paper, the baseline methods resort to external resources in different ways to enhance the base model’s capabilities for the RAG scenario. We compare our proposed Speculative RAG with the baseline method in terms of the external resources as follows.
>
> |                    | Mistral-Instruct; Mixtral-Instruct                   | Self-RAG; Self-CRAG | Speculative RAG                                                                   |
> | ------------------ | ---------- | ------------------ | ----- |
> | External Resources | Extensive Instruction-following data                 | Instruction-following datasets with retrieved documents and synthesized self-reflection tags | Instruction-following datasets with retrieved documents and synthesized rationale |
> | Source             | Huggingface or other in-house SFT or preference data | Curated by GPT-4                                                                             | Curated by Gemini-Ultra (see below for the results with GPT models)       |
> | Size               | About 381,000 (estimated according to OLMo)                  | 145,619 instances                                                                            | 40,059 instances                                                                  |
>
> We list the following new task settings:
>
> **(1) Also using synthesized data from GPT models as the baseline models**
>
> We’ve conducted experiments with GPT for data synthesis, detailed in Appendix A in the submission. We observed that Speculative RAG consistently outperforms all baselines. We are happy to relocate these results to the main text in the final version for better clarity.
>
> |                                               | Trivial QA | PubHealth |
> | --------------------------------------------- | ---------- | --------- |
> | CRAG                                          | 59.03      | 59.04     |
> | Self-RAG                                      | 64.84      | 72.44     |
> | Self-CRAG                                     | 65.43      | 72.85     |
> | Speculative RAG: Drafter-7B + Verification-7B | 72.24      | 73.05     |
>
> **(2) Fine-tuning Mistral on the same synthesized data as our Speculative RAG**
>
> We further tune Mistral-7B with the same synthesized data from Gemini as our Speculative RAG but conduct the standard RAG. We can see our Speculative RAG maintains its performance advantage.
>
> |                                               | Trivial QA | PubHealth | ARC-C |
> | - | - | - | ----- |
> | Mistral-7B                                    | 54.15      | 34.85     | 42.75 |
> | Mistral-7B-Instruct                           | 67.11      | 42.15     | 47.70 |
> | Mistral-7B (fine-tuned)                       | 73.23      | 65.25     | 69.20 |
> | Speculative RAG: Drafter-7B + Verification-7B | 73.91      | 75.79     | 76.19 |
>
> ---
> ## The memory overhead in the parallel verification
>
> Yes, verification is conducted in parallel. We will clarify this in the revised draft. This design choice enables batched input for parallel verification, converting sequentially retrieved documents in the prompt. As shown in Figure 2, generated rationales are significantly shorter than retrieved documents, mitigating the memory burden during parallel verification.
>
>
> ---
> ## Comparison with InstructRAG
>
> We appreciate your bringing InstructRAG to our attention. It is indeed a relevant and noteworthy concurrent work, which we will incorporate into our related work discussion. The use of rationale in instruction-tuning seems to be an independent discovery in both of our works.  Here, we compare Speculative RAG and InstructRAG in detail.
>
> **(1) The Use of Rationale**
>
> InstructRAG and SpeculativeRAG share the use of rationales in instruction-tuning. InstructRAG focuses on denoising retrieved documents and justifying answers, while Speculative RAG leverages smaller and larger language models to balance efficiency and accuracy. The rationale in Speculative RAG primarily serves as a concise reference for verification. We believe these methods are complementary, and incorporating denoising-style rationales into Speculative RAG could further enhance performance.
>
> **(2) Different Problem Setting**
>
> A major difference between InstructRAG and SpeculativeRAG is the problem settings. InstructRAG instruction-tuned the base model with the supervised data from the target dataset while Speculative RAG follows the problem setting of Self-RAG and CRAG and fine-tunes the base model with the general instruction-following dataset. We believe the performance of Speculative RAG can be further improved by incorporating task-specific training data into the instruction-tuning stage.

---

> > ### Author Response · Authors · 2024-11-25
> >
> > Dear reviewer GVM9, we're hoping you might get a chance to check out the rebuttal before the deadline. We'd love to get your feedback!

---

> ### Comment · Reviewer_GVM9 · 2024-12-02
> **Response to Author Rebuttals**
>
> Thank the authors for their response and additional results! My concerns under ```Weaknesses``` are partially address; however, to the best of my knowledge, the two questions I raised under ```Questions``` are not answered by the authors. Collectively, I will maintain the score of my original rating.

---

> ### Author Response · Authors · 2024-12-02
>
> We sincerely thank Reviewer GVM9 for the thoughtful feedback and valuable questions. We have addressed the concerns raised and provided detailed clarifications below. We kindly ask the reviewer to revisit our responses and consider our submission. We are also open to further discussions to improve our work.
>
> > For the comparison of the average number of tokens in the generated rationale and the retrieved documents in Figure 2: What is the number of retrieved documents? Is it before or after subsetting to drafters?
>
> **Answer:** In Figure 2, the number of retrieved documents refers to the subset of documents submitted to the drafter. As described in Section 4.2 (Experiment Settings), this includes 2 documents for TriviaQA, PopQA, PubHealth, and ARC-Challenge experiments, and 6 documents for MuSiQue experiments.
>
> ---
> > The contribution and result of the paper could be further strengthened if newer generation of models (e.g., gemma-2, llama-3) are adopted in the experiments.
>
> **Answer:** We thank the reviewer for raising this great point! Below is the performance analysis of Speculative RAG using the instruction-tuned Gemma-2-2b as the drafter and the frozen Mixtral-8x7B as the verifier:
>
> | Speculative RAG | TQA   | PubHealth | ARC-C |
> | --------------- | ----- | --------- | ----- |
> | Drafter-Mistral-7B + Verification-8x7B |74.24 | 76.60  | 80.55 |
> | Drafter-Gemma-2-2B + Verification-8x7B | 67.22 | 72.14     | 67.92 |
>
> These results suggest that Gemma-2-2b provides a promising avenue for future work. We believe this aligns with the reviewer’s suggestion to explore the capabilities of newer-generation models and opens up opportunities for further optimization.
>
> ---
> > Is the Multi-Perspective Sampling stage also considered in the latency analysis in Section 4.5?
>
> **Answer:** Yes, the latency analysis in Section 4.5 includes the time cost of the Multi-Perspective Sampling stage.

---

> > ### Author Response · Authors · 2024-12-03
> >
> > Dear reviewer GVM9, thank you for your thoughtful feedback. We have updated our responses to address your questions in more detail and hope you have a chance to review them before today's deadline. We sincerely value your insights and look forward to any additional feedback you may have.

---

### Meta-Review · Area_Chair_EHsj · 2024-12-24

**Metareview:**

The paper proposed Speculative RAG, a framework that decomposes retrieval-augmented generation into two stages. First, a specialist draft model produces multiple candidate responses from distinct document subsets. Then, a larger generalist LM verifies these drafts to select a final answer. The authors claim this two-step process (1) improves latency by offloading most of the generation onto a smaller model, and (2) increases accuracy by ensembling generations on diverse retrieval subsets and reducing position bias in lengthy contexts. Experimental evaluations on multiple QA benchmarks show accuracy gains and latency reduction compared to baseline RAG methods.

Strengths
1. The idea paper of adapting speculative decoding into a RAG setting on different retrieval subsets is reasonable.
2. The proposed method achieves improvement over baseline RAG methods on multiple QA benchmarks, with noticable reductions in inference time.
3. The authors conduct thorough ablations to isolate the contributions of different components

Weaknesses
1. Multiple reviewers noted the lack of direct baselines against newly proposed speculative decoding or multi-stage RAG approaches
2. A stronger examination of more modern LLMs could strengthen the paper’s contributions.
3. Certain reviewers view the work as a relatively modest extension of prior multi-stage RAG methods.

The paper presents a novel and efficient approach to RAG, with sufficient empirical results and detailed ablations to support its claims. While some questions about novelty and evaluation were proposed by the reviewers, I believe the authors addressed most points during the rebuttal. The novelty is still a major concern, but considering its simplicity and efficacy, I recommend a weak acceptance of this paper.

**Additional Comments On Reviewer Discussion:**

1. Reviewers raised concerns about fairness due to the use of additional external data. The authors clarified comparisons and added results with newer models.
2. Reviewers noted missing baselines. Authors added results for CRAG, Self-CRAG, and other baselines.
3. Reviewers suggested using more diverse datasets. The authors defended their chosen benchmarks as aligned with prior RAG studies and discussed future expansions.

---

### Decision · Program_Chairs · 2025-01-22

Accept (Poster)